# Efficient quantum thermal simulation

Chi-Fang Chen[1,5 ✉], Michael Kastoryano[2,3], Fernando G. S. L. Brandão[1,2] & András Gilyén[4]

Quantum computers promise to tackle quantum simulation problems that are classically intractable[1]. Although a lot of quantum algorithms[2–4] have been developed for simulating quantum dynamics, a general-purpose method for simulating low-temperature quantum phenomena remains unknown. In classical settings, the analogous task of sampling from thermal distributions has been largely addressed by Markov Chain Monte Carlo (MCMC) methods[5,6]. Here we propose an efficient quantum algorithm for thermal simulation that—akin to MCMC methods—exhibits detailed balance, respects locality and serves as a toy model for thermalization in open quantum systems. The enduring impact of MCMC methods suggests that our new construction may play an equally important part in quantum computing and applications in the physical sciences and beyond.

The equations governing quantum many-body physics are too complex to be solved analytically. Therefore, an arsenal of computational methods—including density functional theory[7], tensor network representations[8,9], and quantum Monte Carlo techniques[10]—has been developed for understanding low-temperature phenomena, ranging from phase transitions and reaction times to ground and thermal state equilibrium properties. However, these classical methods often break down in the presence of strong quantum correlations, for example, exhibited by heavy nuclei, high-temperature superconductors and catalysts.

Future quantum computers promise to solve quantum many-body problems that are classically intractable[1]. At present, a variety of increasingly sophisticated quantum algorithms have been developed for simulating quantum dynamics[2,11,12]. By contrast, for equilibrium properties, the search for particular physical instances in which quantum computers can substantially outperform classical methods remains an area of debate. The efficiency of most existing quantum algorithmic proposals for low-temperature physics[13–16] relies on additional assumptions, which have been challenged in numerical studies[17].

Classically, the many-body thermal simulation problem has been thoroughly addressed, largely by Markov Chain Monte Carlo methods (MCMC). Across a wide range of systems, Metropolis sampling[5] and Glauber dynamics[6] have been successful because of two key features. First, they satisfy detailed balance, which simultaneously lets us (1) prescribe a desired stationary distribution and (2) analyse their convergence (that is, mixing times and spectral gaps). Second, MCMC methods are often local algorithms, that is, the update rule depends only on very few particles at a time; locality gives not only efficient algorithms but also physical connections to how nature thermalizes in a heat bath.

However, quantum computing still lacks a definitive counterpart to MCMC algorithms, as existing proposals[18–21] are nonlocal, break detailed balance, lack conceptual simplicity or require additional hard-to-verify assumptions. Moreover, the algorithm and its analysis in the promising early approach[18] turns out to be incomplete (see Appendix H of ref. 22). As a consequence, progress in quantum thermal simulation remained limited for decades, raising the question whether exact quantum detailed balance should be compatible with the time–energy uncertainty principle—the primary source of difficulties rooted in quantum many-body Hamiltonians with noncommuting terms.

Here, we settle the open problem by giving an efficient quantum algorithm for quantum thermal simulation that satisfies quantum detailed balance exactly while also inheriting the locality of the Hamiltonian, up to a range depending on the temperature. Moreover, our algorithm resembles effective descriptions of open quantum system dynamics when coupling a system to a heat bath. Qualitatively, we expect our algorithm to serve as a minimal, self-contained toy model of thermalization: converging rapidly whenever a system thermalizes in nature and capturing what may physically happen in the case of slow mixing. We speculate that our algorithm might be among the first useful quantum simulation algorithms for low-temperature properties in the forthcoming fault-tolerant era of quantum computing. Furthermore, given the key role played by MCMC algorithms in problems across bioinformatics, social sciences, finance, Bayesian inference and machine learning, we expect an efficient and reliable quantum MCMC algorithm to be a cornerstone of quantum algorithmic development and quantum applications in physical sciences and beyond.

## Quantum MCMC by master equations for thermalization

Our algorithmic solution to thermal simulation hinges on insights into how physical systems thermalize when coupled to a heat bath. In essence, we introduce a conceptually clear, physically motivated toy model of the thermalization process of nature that is also algorithmically efficient. Although early proposals for simulating thermalization directly considered simulating the global system–bath Hamiltonian evolution[23], more tractable approaches use a master equation, or Lindbladian, to capture the effect of a large bath on the small system using a continuous-time Markovian process[24,25]. However, previous approaches inherited issues of the prototypical Davies generator[26,27], which worked well in quantum optics but has unphysical features in noncommuting many-body systems because of the exponentially small level spacing. Our nature-inspired algorithm takes precisely the form of a Lindbladian, inherits the locality from the physical model, exactly satisfies detailed balance and resembles the interactions we expect from weak coupling to a Markovian thermal bath.

[1]Institute for Quantum Information and Matter, California Institute of Technology, Pasadena, CA, USA. [2]AWS Center for Quantum Computing, Pasadena, CA, USA. [3]Department of Computer Science, University of Copenhagen, Copenhagen, Denmark. [4]Lendület "Momentum" Quantum Computing Research Group, HUN-REN Alfréd Rényi Institute of Mathematics, Budapest, Hungary. [5]Present address: Department of EECS, University of California, Berkeley, CA, USA. ✉e-mail: achifchen@gmail.com

**Continuous-time Markov chains and detailed balance.** We begin by reviewing classical MCMC methods, in particular, continuous-time Markov chains defined using the rate matrix $L$ describing infinitesimal transitions on probability distributions p($t$)

$$\frac{dp(t)}{dt} = Lp(t), \tag{1}$$

where $L$ satisfies that $L_{s's} \geq 0$ for each pair of distinct configurations $s \neq s'$ and $L_{ss} = -\sum_{s' \neq s} L_{s's}$ for each configuration $s$. These structural properties ensure that the time evolution can be described by the stochastic matrix p($t$) = $e^{Lt}p(0)$.

The detailed balance condition plays an important part in MCMC methods, requiring that the probability mass transfer between configuration pairs $s, s'$ is symmetric with respect to a target distribution $\pi$

$$L_{s's}\pi_s = L_{ss'}\pi_{s'}, \text{ which then implies stationarity } L\pi = 0. \tag{2}$$

If the target distribution $\pi$ is the unique stationary state, we say the dynamics $e^{Lt}$ is ergodic, and the algorithmic cost to sample from the stationary state is the simulation cost (per unit time $t$) multiplied by the convergence time (the mixing time) of the process.

Given an energy function $H(s)$ of the configurations $s$, an inverse temperature $\beta \geq 0$, and any real nonnegative symmetric transition rate matrix $A_{s's}$, Glauber dynamics[6] is a specific continuous-time Markov chain satisfying detailed balance such that $L_{s's} = \gamma_{s's}A_{s's} - \delta_{s's}\sum_{s''}\gamma_{s''s}A_{s''s}$, where $\gamma_{s's} := \frac{1}{1+e^{\beta(H(s')-H(s))}}$. The process $e^{Lt}$ can be efficiently simulated as long as the energy differences $H(s') - H(s)$ can be computed efficiently for any pair of configurations $s', s$. For example, in classical Ising models, the Hamiltonian is a sum over local interactions, and the energy difference $H(s') - H(s)$ can be computed locally (for transitions that only flip a few neighbouring spins at a time), leading to particularly simple update rules and a deep analytic understanding of mixing times. The robustness and simplicity of Glauber dynamics have led to a comprehensive understanding of phase transitions and thermal properties of classical magnets[28].

**Lindbladians and quantum detailed balance.** The natural quantum counterpart, which arises in the study of open quantum systems, is a master equation, a continuous-time quantum Markov chain, or Lindbladian described by a set of Lindblad operators $\{L_j\}$ and a coherent term $C$:

$$\frac{d\rho(t)}{dt} = \mathcal{L}[\rho(t)] := -\underbrace{i[C, \rho(t)]}_{\text{``coherent''}} + \sum_j \underbrace{L_j\rho(t)L_j^\dagger}_{\text{``transition''}} \\ -\underbrace{\frac{1}{2}(L_j^\dagger L_j\rho(t) + \rho(t)L_j^\dagger L_j)}_{\text{``decay''}}. \tag{3}$$

The above generic form is necessary and sufficient for the evolution $e^{\mathcal{L}t}$ at every $t \geq 0$ to be a valid quantum Markov chain; that is, a completely positive and trace-preserving map. Unlike in the classical case, quantum mechanics allows an additional Hermitian coherent term $C = C^\dagger$ corresponding to the closed-system Schrödinger equation.

Consider an arbitrary noncommuting Hamiltonian with eigen decomposition $H = \sum_i E_i|\psi_i\rangle\langle\psi_i|$. The quantum detailed balance condition with respect to the Gibbs state $\rho_\beta \propto e^{-\beta H}$ requires

$$\langle\psi_2'|\mathcal{L}[|\psi_2\rangle\langle\psi_1|]|\psi_1'\rangle = \exp\left(-\beta\frac{E_1'-E_1+E_2'-E_2}{2}\right)(\langle\psi_2|\mathcal{L}[|\psi_2'\rangle\langle\psi_1'|]|\psi_1\rangle)^* \tag{4}$$

$$\text{for each } |\psi_1\rangle, |\psi_1'\rangle, |\psi_2\rangle, |\psi_2'\rangle,$$

which ensures exact stationarity $\mathcal{L}[\rho_\beta] = 0$ such that $e^{\mathcal{L}t}[\rho_\beta] = \rho_\beta$ for all $t \geq 0$ (refs. 18,29). The above reduces to classical detailed balance (equation (2)) when the states are diagonal in the energy basis (that is, $|\psi_1'\rangle = |\psi_2'\rangle$ and $|\psi_1\rangle = |\psi_2\rangle$). In general, the input quantum state can be a mixture of superpositions of different energy eigenstates, and quantum detailed balance (equation (4)) (specifically, the Kubo–Martin–Schwinger (KMS) detailed balance) requires the transition amplitudes between matrix elements to be related by a Boltzmann factor corresponding to the average of the energy differences $E_1' - E_1$ and $E_2' - E_2$ (Fig. 1).

**Davies generators, the prototype of quantum detailed balance.** In 1974, Davies[26] rigorously derived a Lindbladian from first principles by considering a physical system interacting with a large thermal bath in the weak-coupling and infinite-time limit. Suppose that $A$ is a Hermitian jump operator through which the system is perturbed by the bath (for example, a local Pauli operator); then, the corresponding Lindblad operators $L_\nu$ in the Davies generator are given by a certain Fourier transform of the Heisenberg dynamics

$$L_\nu = \sqrt{\gamma(\nu)}A_\nu, \text{ where } A_\nu := \sum_{i,j:E_i-E_j=\nu}\langle\psi_i|A|\psi_j\rangle|\psi_i\rangle\langle\psi_j| \\ = \lim_{T\to\infty}\frac{1}{2T}\int_{-T}^T e^{iHt}Ae^{-iHt}e^{-i\nu t}dt. \tag{5}$$

These Lindblad operators $L_\nu$ are labelled by the Bohr frequencies $\nu = E_i - E_j \in B(H)$, that is, energy differences between any two eigenstates of the Hamiltonian. Colloquially, $A_\nu$ contains 'the transition amplitudes in $A$ that change the energy by $\nu$', which is precisely what the above operator Fourier transform extracts. The transition weights $\gamma(\nu)$ satisfy the relation $\gamma(\nu)e^{\beta\nu} = \gamma(-\nu)$, whereas their exact form depends on the relaxation dynamics of the bath (see section 'Comparison with physically derived master equations', Methods). The transition weights $\gamma(\nu)$ favour cooling (where energy decreases, $\nu < 0$) and exponentially suppress heating (where energy increases, $\nu > 0$). For diagonal input states and jumps $A$ that preserve diagonal states, taking $\gamma(\nu) = 1/(1 + e^{\beta\nu})$ or $\gamma(\nu) = e^{-\max(\beta\nu,0)}$, the Davies generator recovers the classical Glauber or Metropolis dynamics, respectively.

**Issues with the Davies generator and previous works.** The Davies generator fails to be the definite quantum analogue of Glauber dynamics: while satisfying quantum detailed balance and thus enjoying the desirable mathematical properties, it cannot be implemented efficiently for general many-body systems. The crucial problem is the infinite time integral (equation (5)), which is necessary, because of energy–time uncertainties, for identifying the exact energy difference between eigenstates. The derivation of the Davies generator intends to capture an infinite-time weak-coupling system–bath dynamics and, therefore, has been considered only physical for models in which the integration time can be effectively reduced because of large structured gaps in the spectrum (such as single-particle or commuting Hamiltonians). Although it is possible to truncate the time integral, all existing schemes[18–21] fail to satisfy detailed balance, and some also lead to highly nonlocal Lindblad operators.

## Our new Lindbladian satisfying quantum detailed balance

Now, we construct our Lindbladian that generates a continuous-time quantum Markov chain featuring the key properties of Glauber and Metropolis dynamics: detailed balance and locality. We can view our proposed Lindblad operators as smoothened versions of those from Davies's $L_\nu = \sqrt{\gamma(\nu)}A_\nu$ (equation (5)).

The key idea is to replace the nonphysical transition amplitudes $A_\nu$, which refer to exact energies, with a smooth quantum operator Fourier transform,

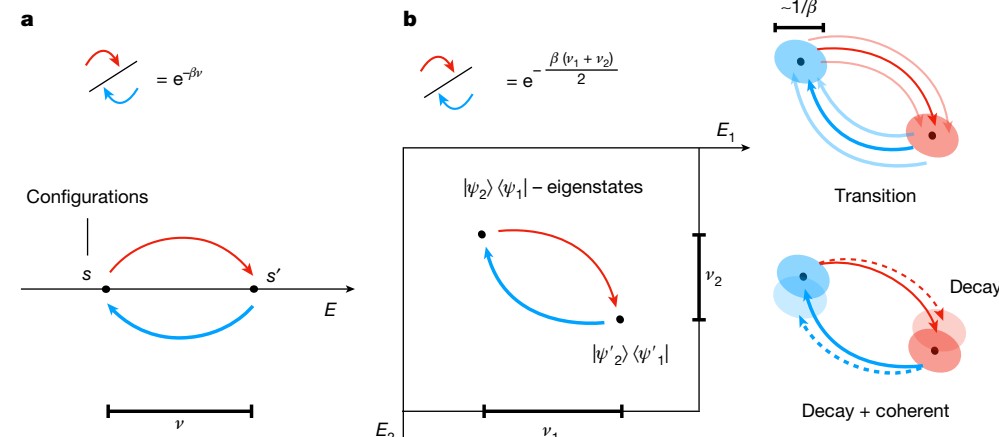

**Fig. 1 | Illustration of detailed balance. a**, Classical: for classical Gibbs distributions, the detailed balance condition is a pairwise relation between heating (red, $v > 0$) and cooling (blue, $v < 0$) transition rates, depending on the energy difference $v$ between states. **b**, Quantum: for quantum Gibbs states, the detailed balance condition refers to matrix elements of the density operator in the energy eigenbasis. Therefore, the relation depends on the respective energy differences $v_1$ and $v_2$ of the corresponding 'kets' and 'bras'. In classical Markov chains, detailed balance can be achieved by rejecting a jump with a probability proportional to the energy difference. However, two key obstacles

arise in implementing quantum detailed balance: (i) measuring energy difference precisely is algorithmically expensive and may disturb the state and (ii) undoing a jump on a quantum state is intricate as we do not have a priori knowledge of the quantum state beyond its energies, and we cannot keep a copy of quantum states before the jump because of quantum no-cloning thereoms. Our construction circumvents these obstacles by smoothly manipulating the transition amplitudes using operator Fourier transforms with energy uncertainty of about $1/\beta$ and introducing a Lindbladian with a corrective coherent term.

$$
\begin{aligned}
\hat{A}(\omega) &:= \sqrt{\frac{\beta}{\sqrt{2\pi}}} \sum_{v \in B(H)} e^{-\beta^2(\omega - v)^2/4} A_v \\
&= \sqrt{\frac{\beta}{\pi\sqrt{2\pi}}} \int_{-\infty}^{\infty} e^{-t^2} e^{-i\beta\omega t} e^{i\beta H t} A e^{-i\beta H t} dt,
\end{aligned}
\tag{6}
$$

which enjoys the sought properties. First, as $\hat{A}(\omega)$ is given by a Gaussian-damped integral of Heisenberg evolutions of $A$, it approximately preserves the locality of the jump $A$ and the Hamiltonian $H$ up to a radius $\beta$ (by Lieb–Robinson bounds, see equation (21) and ref. 30). This (quasi-)locality of $\hat{A}(\omega)$ reflects that detailed balance only depends on the energy difference before and after the jump rather than global energies. Second, the operator $\hat{A}(\omega)$ can be naturally implemented by a phase estimation for operators that selects the transition amplitudes of $A$ that change the energy by roughly $\omega$ (Fig. 3). Technically, equation (6) corresponds to a new variant of quantum phase estimation with a guaranteed Gaussian error profile, which has already found independent applications[31].

Our Lindbladian $\mathcal{L} = \sum_{a \in A} \mathcal{L}^a$ consists of the following carefully constructed Lindbladian terms $\mathcal{L}^a$ associated with the corresponding Hermitian jump operator $A^a$:

$$
\mathcal{L}^a[\rho] := \underbrace{-i[C^a, \rho]}_{\text{"coherent"}} + \underbrace{\int_{-\infty}^{\infty} \gamma(\omega) \hat{A}^a(\omega) \rho \hat{A}^a(\omega)^\dagger d\omega}_{\text{"transition"}} - \underbrace{\frac{1}{2}(D^a \rho + \rho D^a)}_{\text{"decay"}},
\tag{7}
$$

where $\gamma(\omega) := \exp\left(-\max\left(\beta\omega + \frac{1}{2}, 0\right)\right)$,

$$
\begin{aligned}
D^a &:= \int_{-\infty}^{\infty} \gamma(\omega) \hat{A}^a(\omega)^\dagger \hat{A}^a(\omega) d\omega, \text{ and} \\
C^a &:= \int_{-\infty}^{\infty} \frac{i}{\sinh(2\pi t)} (e^{i\beta H t} D^a e^{-i\beta H t} - D^a) dt.
\end{aligned}
\tag{8}
$$

A curious feature of Gaussian energy uncertainty is that the above shifted Metropolis transition weights $\gamma(\omega)$ achieve exact detailed balance for the transition part (equation (15)) despite the presence of energy uncertainty—generalizing the analogous classical case[32]. To fully achieve detailed balance, an extra coherent term $C^a$ is necessary

to accompany the decay term $D^a$, if it does not commute with the system Hamiltonian $H$. It is intriguing that the physical derivation of the Davies generator (equation (5)) also introduces a Lamb-shift term that somewhat resembles our coherent term. However, that Lamb-shift term does not play a part in the stationarity of the Gibbs state because Davies's decay term already satisfies detailed balance as it commutes with $H$ (Methods, Lemma 1).

Finally, note that by directly working with Lindbladians, similarly to ref. 20, we circumvent difficulties around implementing the usual MCMC reject step that arose in earlier approaches because of the no-cloning theorem[18]. See the Methods for more details about our construction and its implementation.

**Simulation costs and guarantees**

As inputs to our algorithm, we require access to a set of Hermitian jumps $\{A^a : a \in A\}$, which is analogous to the symmetric transition rate matrix in the classical case. The jump operators $A^a$ can be as simple as local Pauli operators, but more complex jumps may give faster mixing (for example, cluster updates in Ising models). Our construction can be extended to non-Hermitian jumps, as long as their adjoints are contained in the set[33], but we restrict to Hermitian jumps for simplicity throughout this paper. We quantify the algorithmic cost by the required Hamiltonian simulation time, a subroutine that has been thoroughly studied. The $\widetilde{\mathcal{O}}(\cdot)$ notation absorbs logarithmic dependencies on the evolution time $t$, inverse temperature $\beta$, the strength of the Hamiltonian $\|H\|$, the number of qubits $n$, the inverse trace distance error $1/\epsilon$ and the number of jump operators $|A|$.

**Theorem 1: efficiency, locality and exact detailed balance.** For any $\beta > 0$, Hamiltonian $H$ on $n$-qubits, and self-adjoint jump operators $A^a = A^{a\dagger}$ normalized by $\|\sum_a (A^a)^2\| \le 1$, the Lindbladian $\mathcal{L} = \sum_{a \in A} \mathcal{L}^a$ satisfies the following:

- Each $\mathcal{L}^a$ satisfies quantum detailed balance (equation (4)), and the Gibbs state is stationary $\mathcal{L}^a[\rho_\beta] = 0$.
- For any time $t \ge 1$, the Lindbladian evolution $e^{\mathcal{L}t}$ can be implemented up to $\epsilon$-error (in diamond distance), using

$$\pm \widetilde{\mathcal{O}}(\beta \cdot t) \text{ Hamiltonian simulation time,}$$

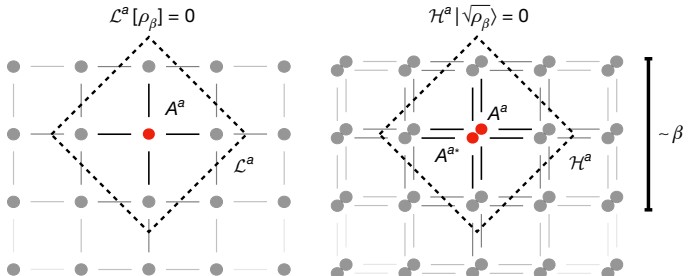

**Fig. 2 | Quasi-locality of the maps.** For a lattice Hamiltonian, our Lindbladian is a sum over quasi-local terms $\mathcal{L}^a$ surrounding each jump $A^a$ with radius approximately $\beta$, with an exponentially decaying tail controlled by the Lieb–Robinson bounds. This locality remains to hold for the parent Hamiltonian $\mathcal{H} = \sum_a \mathcal{H}^a$ of the purified Gibbs state $|\sqrt{\rho_\beta}\rangle$ defined on two copies of the system.

$\widetilde{\mathcal{O}}(t)$ applications of the block encoding (and its inverse) for the jumps $\sum_{a \in A} |a\rangle \otimes A^a$, $\widetilde{\mathcal{O}}(t)$ two-qubit gates and (apart from those in the block encoding and the Hamiltonian simulation) $\widetilde{\mathcal{O}}(1)$ ancillas.

- If the Hamiltonian $H$ is geometrically local, then each summand $\mathcal{L}^a$ is well-approximated by a local Lindbladian acting on $\widetilde{\mathcal{O}}(\beta)$-radius ball surrounding the jump operator $A^a$.
- If the set of jump operators $\{A^a\}$ has no nontrivial invariant subspace (for example, all single-site Pauli $X$ and $Z$ operators), then the Gibbs state $\rho_\beta$ is the unique fixed point.

Detailed balance has been a central feature in the study of Markov chain mixing time. The lack of exact quantum detailed balance, mainly rooted in imprecise energy measurements, has been an obstacle for noncommuting Hamiltonians. Here, achieving exact quantum detailed balance makes a formal connection to the existing framework and enables clean mathematical reasoning, as demonstrated by a series of recent works[34–36] that used our Lindbladian to prove rapid mixing for lattice Hamiltonians at high temperatures.

Detailed balance also enables mapping our Lindbladian to a frustration-free parent Hamiltonian $\mathcal{H}$ acting on a doubled, purifying system[19,20,37], whose zero-eigenstate is a certain purified Gibbs state $|\sqrt{\rho_\beta}\rangle$ (known as the thermal field double in high-energy physics) (Fig. 2). This gives a coherent Gibbs sampler, in which we prepare the purified Gibbs state by a natural adiabatic path parametrized by the inverse temperature $\beta$. Of course, the purified Gibbs state reduces to the (mixed) Gibbs state once we trace out the purifying replica system, but the purification opens doors to advanced quantum algorithmic tools, such as verification (for example, through the swap test or measuring the energy of the parent Hamiltonian) or faster mean estimation[38].

The key to the efficiency of our proposal lies in the revelation that detailed balance generally does not require high-precision metrology. The Hamiltonian simulation time ~$\beta$ (suppressing other dependencies) used in our Lindbladian is not enough for any quantum algorithm to learn energies beyond precision ~$1/\beta$. Yet, it is sufficient for coherently modulating the transition amplitudes according to the quantum detailed balance condition (equation (4)), leading to improved implementation methods that are analogous to the exponentially more precise quantum linear equation solvers[39], which circumvent the metrology step of the original phase-estimation-based proposal[40].

We can further exploit that the modulation depends only on the energy differences before and after jumps. Thus, for geometrically local Hamiltonians, our Lindbladian diagnoses energy differences by (quasi-)localized Hamiltonian patches of radius ~$\beta$, which, in turn, could further improve the efficiency of implementation by truncating faraway Hamiltonian terms. For classical or commuting Hamiltonians, the energy difference depends only on a strictly local neighbourhood, regardless of $\beta$, which simplifies the corresponding Gibbs samplers. This aspect of locality contrasts with earlier quantum methods, which

typically involve global energy measurements, for example, through phase estimation.

To implement our synthetic Lindbladian, we also develop high-precision Lindbladian simulation algorithms (Theorem 2), giving an explicit quantum Gibbs sampler with provable accuracy guarantees. However, from a practical perspective, we expect there to be opportunities for heuristic optimization of the actual circuit implementation, for example, by simulating the Lindblad dynamics to low orders (see Fig. 3 for a first-order implementation), truncating the Lindbladian operator depending on the state, or tuning the discretization parameters.

## Mixing times

We have shown that given a local Hamiltonian $H$, our new construction allows for a step-wise efficient quantum MCMC algorithm. However, just as in the classical case, the mixing times can depend crucially on the specific problem at hand. Although large-scale numerical exploration of mixing times for strongly interacting systems seems to be intractable without fault-tolerant quantum computers, we can exactly diagonalize the Lindbladian for small systems, which can, for example, reveal interesting phenomena related to the choice of jump operators[41]. For illustration, we choose to look at the transverse field Ising model $H_{\text{TFI}} = -\sum_j Z_j Z_{j+1} + \lambda X_j$ and the XXZ model $H_{\text{XXZ}} = \sum_j X_j X_{j+1} + Y_j Y_{j+1} + \gamma Z_j Z_{j+1}$ on a ring. Here, $X$, $Y$ and $Z$ are the single-site Pauli operators, and $\lambda$ and $\gamma$ are the parameters of the models. Neither model exhibits a phase transition in its thermal states, as it is on a one-dimensional lattice[42]. Hence, we expect a system-size-independent gap in the spectrum of the Lindbladian at all temperatures (when each jump is normalized to have unit strength $\|A^a\| = 1$). By contrast, the models show quantum phase transitions in their ground states.

In Fig. 4, we simulate the two models for system size $n = 8$ by exactly diagonalizing the Lindbladian in its matrix representation (diagonalizing larger systems is prohibitive because the number of matrix elements grows as $2^{4n}$). At high temperatures ($\beta \ll 1$), we observe a large spectral gap in all parameter regimes for both models with the Lindbladian constructed from local Pauli $X$, $Y$ and $Z$ operators only, in agreement with the recent theoretical analysis[34,35]. At low temperatures, both models exhibit a critical slowdown at certain values of the parameters $\lambda$ and $\gamma$. However, adding a global jump $X^{\otimes n} = \prod_j X_j$ lifts the critical slowdown in the ferromagnetic phases, while adding two-body terms $\{X_j X_{j+1}\}$ lifts the gap in the antiferromagnetic phase of the XXZ model. This is consistent with classical Glauber dynamics, in which slow mixing of the two-dimensional Ising model at low temperatures can be resolved by applying global flips. We take this as positive preliminary evidence that our quantum Gibbs samplers are consistent and can be expected to allow for model-specific heuristic improvements, similar to classical MCMC.

## Future directions and applications

A central challenge in quantum simulation is the preparation of thermal states. For this purpose, we have constructed a physically inspired, algorithmically efficient and theoretically clean Lindbladian that matches the celebrated features of classical MCMC algorithms. Our new construction opens up several avenues for further studies:

- A provable quantum simulation algorithm: the study of spectral gaps and mixing times of classical Markov chains has been a fruitful and persistent subject[28,43,44]. Our construction gives a well-defined, quasi-local map with exact detailed balance and invites a systematic study for noncommuting Hamiltonians. Recently, rapid mixing at perturbative regimes or high temperatures has been proved[34–36] using our Lindbladians; through a dedicated mixing time analysis, refs. 34,45 show that closely related cooling processes at low temperatures are universal for quantum computation, suggesting that simulating low-temperature cooling dynamics can be classically hard. Ultimately, directly showing that physically relevant systems (for example, the Fermi–Hubbard model) mix rapidly at classically

**a**

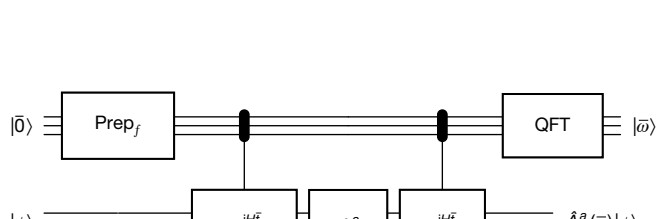

**b**

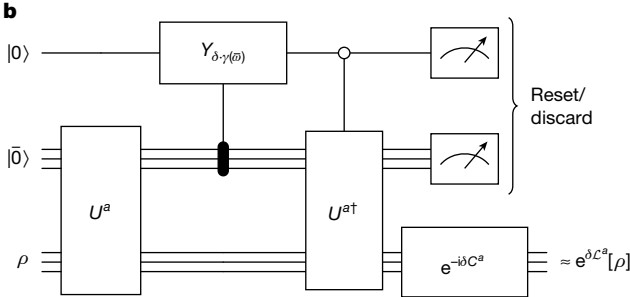

**Fig. 3 | A circuit implementation of the Lindbladian evolution to first-order with error $\mathcal{O}(\delta^2)$. a**, Circuit implementation for the isometry $U^a$ representing the discrete quantum operator Fourier transform $\hat{A}^a(\bar{\omega})$, evaluated on input $|\bar{0}\rangle \otimes |\psi\rangle$, in which we assumed for simplicity that $A^a$ is a Hermitian unitary. The subroutines include Gaussian filter state preparation $|f\rangle$ (equation (6)), controlled-Hamiltonian simulation and quantum Fourier transform. The label $\bar{\omega}$ is a rough estimate of the energy difference before and after $A^a$ (that is, the Bohr frequency). **b**, Quantum circuit implementation of an approximate $\delta$-time step corresponding to a single jump operator $A^a$. This weak-measurement

scheme uses the quantum operator Fourier transform isometry $U^a$ and its inverse, a controlled single-qubit rotation $Y_\theta = \begin{bmatrix} \sqrt{1-\theta} & -\sqrt{\theta} \\ \sqrt{\theta} & \sqrt{1-\theta} \end{bmatrix}$, Hamiltonian simulation of the coherent part $C^a$ and ancilla resets. To simulate the full Lindbladian, this circuit should be repeatedly applied with the jump operator $A^a$ for a uniformly random $a \in A$. We can implement a block encoding of $C^a$ by a linear combination of unitaries[11] according to equation (8). Alternatively, ignoring the $C^a$ term, detailed balance becomes approximate, with error controlled by the precision of energy difference estimation[22].

intractable regimes gives quantitative evidence of practical quantum advantage in quantum simulation.

- A dynamical angle to quantum Gibbs states: the mixing time of a Lindbladian is naturally connected to the correlation decay and complexity of quantum Gibbs states. It was a celebrated result in classical Gibbs sampling that rapid mixing is deeply tied with the decay of Gibbs correlation[28], and only recently has the equivalence been extended to some commuting quantum Hamiltonians[46,47]. Our newly defined map gives a dynamic angle for rigorously studying intricate quantum correlations, such as the area-law of entanglement, conditional mutual information[48–50], and topological order in noncommuting Gibbs states.

- A self-contained toy model of open-system thermodynamics: as a quantum analogue of Glauber dynamics, the qualitative parallel to physical master equations poses our construction as a succinct toy model of open-system physics. Thus, in cases in which we cannot prepare the Gibbs state due to a mixing time slowdown, we may still uncover properties of new thermal phases. Our Lindbladian provides a theoretical starting point for studying metastability, energy

landscapes, quantum spin glasses and self-correcting quantum memories, whose precise formulation for noncommuting Hamiltonians has also been lacking. The explicit form of our Lindbladian also enables direct numerical studies regarding the above notions.

In light of the key theoretical and empirical success of MCMC methods in classical computing, we expect quantum Gibbs sampling to serve similarly important roles in quantum computing. Given the current scepticism about the practical applicability of quantum computers, we anticipate our results will initiate a new set of diverse research directions covering theory, experiment, numerics and applications.

## Online content

**a**

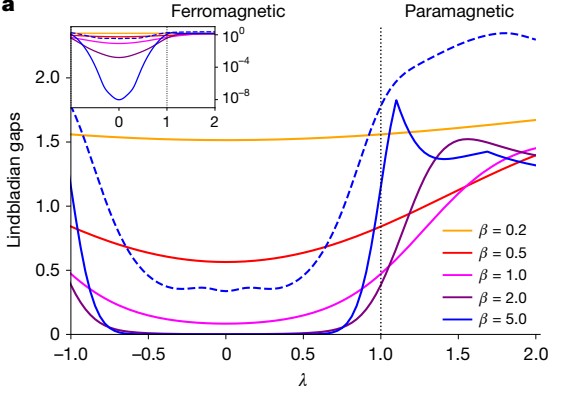

**b**

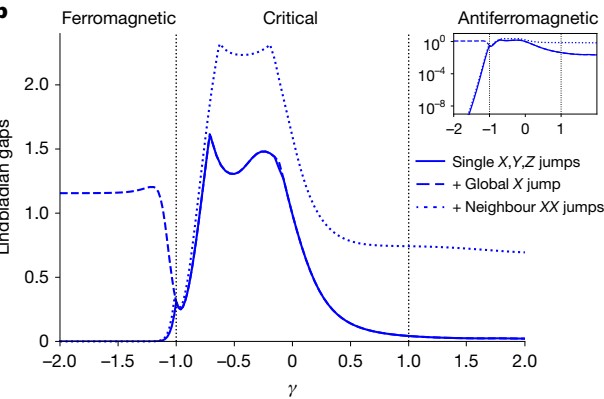

**Fig. 4 | Lindbladian gaps as a function of parameters of two spin chain Hamiltonians. a**, The transverse field Ising model with periodic boundary conditions on $n = 8$ qubits at various temperatures, with single Pauli jumps. As expected, the Lindbladian is gapped at high temperatures throughout the phase diagram, and the gap closes exponentially in $\beta$ in the ferromagnetic regime (which is well-known in the $\lambda = 0$ case from the classical literature). The inset in the top outer corner shows the same gaps on a logarithmic scale. Vertical dotted lines highlight the parameter values at which a quantum

phase transition happens in the ground state of the models, and the phases are labelled above. The dashed line shows that, even for the lowest temperature $\beta = 5$, the Lindbladian gap can be opened by including global spin-flip $X^{\otimes n}$. **b**, The Heisenberg XXZ model with periodic boundary conditions on $n = 8$ qubits at a low temperature $\beta = 5$, with (i) only local Pauli jumps, (ii) local Pauli jumps and a global Pauli $X^{\otimes n}$ jump and (iii) local Pauli jumps and the nearest-neighbour Pauli $XX$ jumps. The dynamics can freeze with single-site jumps, but adding global and nearest-neighbour Pauli $XX$ jumps speeds up mixing by opening the gap.

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

# Methods

We provide the explicit form of our construction and explain how exact detailed balance can be achieved through the key algorithmic subroutines and circuits.

## Constructing the full Lindbladian

Recall that our main result considers the following Lindbladian in the Schrödinger picture:

$$\mathcal{L}[\,\cdot\,] := \sum_{a\in A} \underbrace{-i[C^a,\cdot\,]}_{\text{``coherent''}}$$

$$+ \overbrace{\int_{-\infty}^{\infty} \gamma(\omega)\left(\underbrace{\hat{A}^a(\omega)[\,\cdot\,]\hat{A}^a(\omega)^\dagger}_{\text{``transition''}} - \underbrace{\frac{1}{2}\{\hat{A}^a(\omega)^\dagger\hat{A}^a(\omega),\cdot\,\}}_{\text{``decay''}}\right)\mathrm{d}\omega}^{\text{dissipative part}}. \tag{9}$$

Roughly, it resembles the Davies generator (equation (5)) but is carefully modified to maintain quantum detailed balance while ensuring locality and efficiency. We begin by reviewing the detailed balance condition for the Davies generator to illustrate the key ingredients and differences in our construction. We may regroup the above according to the jumps $\mathcal{L} = \sum_{a\in A}\mathcal{L}^a$, and study each term individually, so in what follows we drop the label $a$ for simplicity, that is, substitute $A^a \leftarrow A$.

**Detailed balance of the Davies generator.** For a Hermitian jump $A$, recall that the Davies generator $\mathcal{L}_{Davies}[\,\cdot\,] := \sum_\nu \gamma(\nu)A_\nu[\,\cdot\,]A_\nu^\dagger - \frac{\gamma(\nu)}{2}\{A_\nu^\dagger A_\nu,\cdot\,\}$ satisfies the KMS-detailed balance condition (equation (4)), that is, satisfies the superoperator equation

$$\mathcal{L}[\,\cdot\,] = \rho^{\frac{1}{2}}\mathcal{L}^\dagger[\,\rho^{-\frac{1}{2}}\cdot\rho^{-\frac{1}{2}}]\rho^{\frac{1}{2}}, \tag{10}$$

where $\mathcal{L}^\dagger$ is the superoperator adjoint of $\mathcal{L}$ defined by $\mathrm{Tr}[(\mathcal{L}[X])^\dagger Y] = \mathrm{Tr}[X^\dagger\mathcal{L}^\dagger[Y]]$ for all $X, Y$. Here, detailed balance hinges on the following exact operator-valued symmetries: for all Bohr frequency $\nu \in B(H)$,

$$\rho^{-\frac{1}{2}}A_\nu\rho^{\frac{1}{2}} = e^{\frac{\beta\nu}{2}}A_\nu \quad\text{(conjugation identity)},$$

$$A_{-\nu} = (A_\nu)^\dagger \quad\text{(adjoint property)}.$$

The conjugation identity is rooted in that $A_\nu$ are eigenoperators $[H, A_\nu] = \nu A_\nu$ of the commutator $[H,\cdot\,]$, highlighting a special role played by the Bohr-frequency decomposition $A_\nu$. The adjoint property says that for a Hermitian jump $A$, the transition amplitudes associated with energy difference $\nu$ are paired with the reverse difference $-\nu$, reminiscent of a Fourier transform symmetry of real functions.

As a consequence, the decay part readily satisfies detailed balance by itself because of the adjoint property, because the operator $(A_\nu)^\dagger A_\nu = A_{-\nu}A_\nu$ in the decay part preserves the energies and commutes with the Hamiltonian (and hence with $\rho$). For the transition part,

$$\sum_\nu \gamma(\nu)\rho^{\frac{1}{2}}(A_\nu)^\dagger\rho^{-\frac{1}{2}}\cdot\rho^{-\frac{1}{2}}A_\nu\rho^{\frac{1}{2}}$$

$$= \sum_\nu \gamma(\nu)e^{\beta\nu}(A_\nu)^\dagger\cdot A_\nu \quad\text{(by the conjugation identity)} \tag{11}$$

$$= \sum_\nu \gamma(-\nu)A_{-\nu}\cdot(A_{-\nu})^\dagger = \sum_\nu \gamma(\nu)A_\nu\cdot(A_\nu)^\dagger.$$

The second equality uses the Kubo–Martin–Schwinger condition (in the frequency domain) for the transition weights

$$\gamma(\nu)e^{\beta\nu} = \gamma(-\nu). \tag{12}$$

The main obstacle towards a Lindbladian with exact detailed balance and efficient implementation is the lack of algorithmic access to $A_\nu$ in the presence of a dense spectrum, as approximations to $A_\nu$ can easily break the exact symmetry conditions in detailed balance[18,20,51].

**Operator Fourier transforms.** The key to both efficiency and exact detailed balance is a careful relaxation of $A_\nu$ that preserves some aspects of the symmetries using the operator Fourier transform damped by a Gaussian filter $f(t) := e^{-\sigma^2 t^2}\sqrt{\sigma\sqrt{2/\pi}}$ (normalized by $\int_{-\infty}^{\infty}|f(t)|^2\mathrm{d}t = 1$) with a tunable width $\propto 1/\sigma$ (setting $\sigma = \frac{1}{\beta}$ recovers (equation (6))):

$$\hat{A}(\omega) := \frac{1}{\sqrt{2\pi}}\int_{-\infty}^{\infty} e^{iHt}Ae^{-iHt}e^{-i\omega t}f(t)\mathrm{d}t$$

$$= \frac{1}{\sqrt{\sigma\sqrt{2\pi}}}\sum_{\nu\in B(H)} A_\nu e^{-\frac{(\omega-\nu)^2}{4\sigma^2}}. \tag{13}$$

Like $A_\nu$ in the Davies generator (equation (5)), our operator Fourier transforms $\hat{A}(\omega)$ are labelled by energy differences, but here the parameter $\omega \in [-\infty, \infty]$ takes continuous values, without referring to the true Bohr frequencies. When the uncertainty vanishes $\sigma \to 0$, we recover $\lim_{\sigma\to 0}\sqrt{\sigma\sqrt{2\pi}}\,\hat{A}(\omega) = \sum_{\nu\in B(H)}\mathbb{1}(\nu = \omega)A_\nu$; when the energy uncertainty is finite $\sigma \neq 0$, the operator Fourier transforms still maintain exact algebraic properties crucial for detailed balance. First, conjugating with the Gibbs state preserves the form of operator Fourier transforms, albeit causing a shift and rescaling

$$\rho^{-\frac{1}{2}}\hat{A}(\omega)\rho^{\frac{1}{2}} = \frac{1}{\sqrt{\sigma\sqrt{2\pi}}}\sum_{\nu\in B(H)} A_\nu e^{\frac{\beta\nu}{2}}\exp\left(-\frac{(\omega-\nu)^2}{4\sigma^2}\right) \quad\text{(by (13)}$$

$$\text{and the conjugation identity)}$$

$$= \frac{1}{\sqrt{\sigma\sqrt{2\pi}}}\sum_{\nu\in B(H)} e^{\frac{\beta\omega}{2}+\frac{\sigma^2\beta^2}{4}}A_\nu\exp\left(-\frac{(\omega-\nu+\sigma^2\beta)^2}{4\sigma^2}\right)$$

$$\text{(by completing the square)}$$

$$= e^{\frac{\beta\omega}{2}+\frac{\sigma^2\beta^2}{4}}\hat{A}(\omega+\sigma^2\beta), \quad\text{(by shift-rescale symmetry)}$$

which reflects the fact that multiplying a Gaussian distribution by an exponential weight must shift the mean $e^{(x-a)^2-2bx} = e^{(x-a-b)^2-2ab-b^2}$. Second, even though the operator Fourier transform is a linear combination of different Bohr frequencies, the adjoint property holds exactly when $A = A^\dagger$ as

$$\hat{A}(-\omega) = \frac{1}{\sqrt{2\pi}}\int_{-\infty}^{\infty} e^{iHt}Ae^{-iHt}e^{i\omega t}f(t)\mathrm{d}t$$

$$= \frac{1}{\sqrt{2\pi}}\int_{-\infty}^{\infty} e^{iHt}A^\dagger e^{-iHt}(e^{-i\omega t}f(t))^*\mathrm{d}t = \hat{A}(\omega)^\dagger \quad\text{(since } f^*(t) = f(t)).$$

The above two exact symmetries appear to be absent in previous approaches that attempted to directly measure energy differences. Now we show that quantum detailed balance is a consequence of these exact symmetries and related algebraic properties of the Gaussian uncertainty in operator Fourier transforms, which hold exactly despite not measuring energies to high precision.

**Exact detailed balance with finite uncertainty.** Although the operator Fourier transform does not yield an exact representation of $A_\nu$, the uncertainty has a specific structure with distinctive symmetries— arising from the interplay between the Gaussian weighing and the exponential form of the Boltzmann factors—and consequently can be exactly compensated by an appropriate shift in the transition weights.

In the following, we prove that the transition part (equation (9)) satisfies detailed balance (equation (10)): $\mathcal{T}[\,\cdot\,]=\rho^{\frac{1}{2}}\mathcal{T}^\dagger[\rho^{-\frac{1}{2}}\cdot\rho^{-\frac{1}{2}}]\rho^{\frac{1}{2}}$. The above shift-rescale and adjoint symmetries yield

$$\rho^{\frac{1}{2}}\mathcal{T}^\dagger[\rho^{-\frac{1}{2}}\cdot\rho^{-\frac{1}{2}}]\rho^{\frac{1}{2}}$$
$$=\int_{-\infty}^\infty \gamma(\omega)\left(\rho^{-\frac{1}{2}}\hat{A}(\omega)\rho^{\frac{1}{2}}\right)^\dagger[\,\cdot\,]\rho^{-\frac{1}{2}}\hat{A}(\omega)\rho^{\frac{1}{2}}\,d\omega$$
(by definition)
$$=\int_{-\infty}^\infty \gamma(\omega)e^{\beta\omega+\frac{\sigma^2\beta^2}{2}}\hat{A}(-\omega-\sigma^2\beta)[\,\cdot\,]\hat{A}(-\omega-\sigma^2\beta)^\dagger\,d\omega,$$
(14)

showing that we may compensate for the uncertainty by imposing a shifted KMS condition (equation (12))

$$\gamma(\omega)e^{\beta\omega+\frac{\sigma^2\beta^2}{2}}=\gamma(-\omega-\sigma^2\beta^2),\ \text{ such that}$$
$$(14)=\int_{-\infty}^\infty \gamma(-\omega-\sigma^2\beta)\hat{A}(-\omega-\sigma^2\beta)[\,\cdot\,]\hat{A}(-\omega-\sigma^2\beta)^\dagger\,d\omega$$
$$=\mathcal{T}[\,\cdot\,].$$
(15)

Therefore, we can take any transition weight function satisfying the KMS condition $\gamma_0(v)e^{\beta v}=\gamma_0(-v)$ and pretend we underestimated the Bohr frequency by $\frac{\sigma^2\beta}{2}$, that is, substitute $v\leftarrow\omega_+:=\omega+\frac{\sigma^2\beta}{2}$ to obtain our shifted $\gamma(\omega)$. The canonical examples include the Metropolis and Glauber weights

$$\gamma^M(\omega):=\exp(-\beta\max(\omega_+,0)).\text{ (shifted Metropolis)}$$
(16)

$$\gamma^G(\omega):=\frac{1}{1+e^{\beta\omega_+}}.\text{ (shifted Glauber)}$$
(17)

See ref. 33 for the original step-by-step detailed derivation; the above streamlined derivation draws partly from subsequent simplification and development (Lemma 7.1 of ref. 52 and Lemma IX.2 of ref. 49).

Under the natural normalization $\gamma(\omega)\in[0,1]$, the shifted symmetry (equation (15)) implies a low transition rate $\gamma(0)\le\exp(-\sigma^2\beta^2/2)$ around $\omega=0$. Therefore, to avoid unnecessary idling of the process, it is imperative to choose an uncertainty $\sigma\le1/\beta$ not exceeding the temperature; in the main text, we have simply set $\sigma=\frac{1}{\beta}$. This is why implementing a single step uses $-\beta$ Hamiltonian simulation time in our construction. By contrast, for classical or commuting systems with gapped periodic spectrum, the uncertainty can be discretized, and there is no need to scale the Hamiltonian simulation time with $\beta$.

**Achieving full detailed balance by tuning the coherent term.** Even if the transition part (equation (9)) satisfies detailed balance exactly, the decay part $D$ of the Lindbladian may still break detailed balance when it does not commute with the Hamiltonian $H$.

A second insight of our construction is to prescribe and efficiently implement a dedicated coherent term $C$ that perfectly cancels out the deviation from quantum detailed balance in a uniquely quantum way, as shown by the following lemma.

**Lemma 1: prescribing the coherent term.** (Lemma II.1 of ref. 33) For any full-rank density operator $0\prec\rho\in\mathbb{C}^{d\times d}$ and Hermitian operator $D\in\mathbb{C}^{d\times d}$, there is a unique Hermitian operator $C\in\mathbb{C}^{d\times d}$ (up to adding any scalar multiples of the identity $I$) such that the superoperator

$$-\frac{1}{2}\{D,\cdot\}-\mathrm{i}[C,\cdot]$$
(18)

satisfies $\rho$-detailed balance. For a Gibbs state $\rho\propto\exp(-\beta H)$, we can express $C$ as

$$C=\sum_{v\in B(H)}\frac{\mathrm{i}}{2}\tanh\left(\frac{\beta v}{4}\right)D_v$$
$$\text{where}\quad D_v:=\sum_{E_i-E_j=v}|\psi_i\rangle\langle\psi_i|D|\psi_j\rangle\langle\psi_j|.$$
(19)

**Proof.** The detailed balance condition is equivalent to the following matrix being self-adjoint:

$$\rho^{-\frac{1}{4}}\left(-\frac{D}{2}-\mathrm{i}C\right)\rho^{\frac{1}{4}}=\frac{1}{2}\rho^{-\frac{1}{4}}\left(\sum_{v\in B(H)}D_v\left(\tanh\left(\frac{\beta v}{4}\right)-1\right)\right)\rho^{\frac{1}{4}}$$
$$=-\frac{1}{2}\sum_{v\in B(H)}\frac{D_v}{\cosh\left(\frac{\beta v}{4}\right)}.$$

This is self-adjoint because $(D_v)^\dagger=D_{-v}$, $\cosh(x)=\cosh(-x)$, and $B(H)=-B(H)$. $\square$

See also ref. 53 on prescribing fixed points in general, without necessarily assuming detailed balance. The coherent term $C$ is a Hermitian matrix obtained by reweighing the given $D$ operator with the profile $\mathrm{i}\tanh(\beta v/4)/2$ on each Bohr frequency component $D_v$. The coherent term is completely determined by $\rho$ and $D$ (ref. 54), and we found a particularly simple and useful closed-form representation in the time domain (recall $\operatorname{sinhc}(x):=\sinh(x)/x$ for real $x\ne0$)

$$C=\int_{-\infty}^\infty\frac{\mathrm{i}}{\sinh(2\pi t)}(e^{\mathrm{i}\beta Ht}De^{-\mathrm{i}\beta Ht}-D)\,dt$$
$$=\frac{\beta}{2\pi}\int_{-\infty}^\infty\frac{-1}{\operatorname{sinhc}(2\pi t)}\int_0^1 e^{\mathrm{i}s\beta Ht}[H,D]e^{-\mathrm{i}s\beta Ht}\,ds\,dt,$$
(20)

which in turn implies the algorithmic efficiency and quasi-locality of the coherent term. The above integral form and the exponential decay in $t$ allow using the linear combination of unitaries technique to implement a block encoding of $C$ (refs. 33,55) by truncating and discretizing the time integral. For the Davies generator (equation (5)), we have that $[H,D]=0$. Therefore, the dissipative part readily satisfies detailed balance, and $D_v=0$ for all $v\ne0$, implying that the above coherent term prescription simply vanishes.

**Quasi-locality.** A salient feature of our construction is the quasi-locality of the Lindbladian terms, inherited from the operator Fourier transform in systems that feature a Lieb–Robinson bound. For all $\omega\in\mathbb{R}$ and geometrically local jump $A$, truncating the Hamiltonian to a local Hamiltonian patch $H_\ell$ within distance $\ell$ from the jump yields an error

$$\left\|\hat{A}_{(H)}(\omega)-\hat{A}_{(H_\ell)}(\omega)\right\|\le\frac{1}{\sqrt{2\pi}}\int_{-\infty}^\infty\|e^{\mathrm{i}Ht}Ae^{-\mathrm{i}Ht}-e^{\mathrm{i}H_\ell t}Ae^{-\mathrm{i}H_\ell t}\|\,|f(t)|\,dt.$$
(21)

For a wide variety of local Hamiltonian systems, off-the-shelf Lieb–Robinson bounds[30] for the Heisenberg dynamics $e^{\mathrm{i}Ht}Ae^{-\mathrm{i}Ht}$ state that the integrand in equation (21) exponentially reduces with the distance $\ell$ but degrades with the evolution time $t$. As the Gaussian weight function $f(t)$ of equation (13) effectively dampens the integral to small values of $t\sim1/\sigma$, the operator $\hat{A}_{(H)}(\omega)$ is well-approximated by a Hamiltonian patch with radius scaling with the energy uncertainty $\sigma$, which can be independent of the system size. See, for example, Appendix A of ref. 49 for a quantitative estimate for both the transition and coherent parts. Of course, after truncation, the resulting strictly local Lindbladian may no longer satisfy exact quantum detailed balance.

**Fixed point and its uniqueness.** The detailed balance condition (equation (10)) directly implies that the Gibbs state $\rho$ is a fixed point for the Lindbladian $\mathcal{L}$.

$$\mathcal{L}[\rho] = \rho^{\frac{1}{2}} \mathcal{L}^{\dagger}\left[\rho^{-\frac{1}{2}}(\rho^{\frac{1}{2}})\rho^{-\frac{1}{2}}\right]\rho^{\frac{1}{2}} = \rho^{\frac{1}{2}}\mathcal{L}^{\dagger}[I]\rho^{\frac{1}{2}} = 0,$$

where the last equality used the trace-preservation property $\mathcal{L}^{\dagger}[I] = 0$ of Lindbladians.

We now explain why the Gibbs state is the unique fixed point whenever the set of jump operators has no (nontrivial) invariant subspaces, which holds, for example, when the jumps include all single-site Pauli $X$ and $Z$ operators. Decomposing each jump operator by the operator Fourier transform $A^a \propto \int_{-\infty}^{\infty} \hat{A}^a(\omega)d\omega$ cannot create new invariant subspaces; likewise, multiplying by strictly positive transition weights $\gamma(\omega)$ cannot. Thus, the resulting set of Lindblad operators $\sqrt{\gamma(\omega)}\hat{A}^a(\omega)$ has no invariant subspaces, which is known[56] to imply the uniqueness of the fixed point. But this uniqueness argument says little about the quantitative convergence rate, and the mixing times can depend on the particular Hamiltonian.

**A single-qubit example.** Let us illustrate the details of our Lindbladian construction through a pedagogical example. Consider a single-qubit Hamiltonian with a single jump:

$$H = Z = \begin{bmatrix} 1 & 0 \\ 0 & -1 \end{bmatrix} = |0\rangle\langle 0| - |1\rangle\langle 1| \quad \text{and}$$

$$A = X = \begin{bmatrix} 0 & 1 \\ 1 & 0 \end{bmatrix} = |1\rangle\langle 0| + |0\rangle\langle 1|. \tag{22}$$

The eigenvalues of the Hamiltonian are $\pm 1$, and the Bohr frequencies, their differences, are $B(H) = \{2, 0, -2\}$. We can decompose the jump by the Bohr frequencies, into the $\nu = 2$ and $\nu = -2$ components as follows:

$$A_2 = |0\rangle\langle 1| \quad \text{and} \quad A_{-2} = |1\rangle\langle 0|. \tag{23}$$

We can then directly obtain the Davies generator for any transition weights satisfying $\gamma(2)e^{2\beta} = \gamma(-2)$. By contrast, our operator Fourier transform yields

$$\hat{A}(\omega) = \frac{1}{\sqrt{\sigma\sqrt{2\pi}}}\left(e^{-\frac{(\omega-2)^2}{4\sigma^2}}A_2 + e^{-\frac{(\omega+2)^2}{4\sigma^2}}A_{-2}\right). \tag{24}$$

Thus, for all $\omega \in \mathbb{R}$ the resulting $\hat{A}(\omega)$ has contributions coming from both the $A_2$ and $A_{-2}$ components; these components only get separated in the $\sigma \to 0$ limit. Nevertheless, detailed balance still holds at every finite uncertainty $\sigma$ with suitable choices of $\gamma(\omega)$ (equation (15)) and the additional coherent term.

**Recovering the Davies generator.** As we show here, our Lindbladian exactly recovers the Davies generator in the $\sigma \to 0$ limit. Moreover, the Davies generator reduces to Glauber dynamics when the Hamiltonian $H$ is a classical function of bitstrings and the jumps $A$ map a bitstring to another: in this case, inputs that are a probabilistic mixture of bitstrings undergo a classical Glauber dynamics (see section 'Quantum MCMC by master equations for thermalization').

For any bounded, continuous function $\gamma_0$ satisfying the KMS condition $\gamma_0(\nu)e^{\beta\nu} = \gamma_0(-\nu)$, consider the shift $\gamma_{\beta\sigma}(\omega) := \gamma_0\left(\omega + \frac{\beta^2\sigma^2}{2}\right)$, then the transition part (similarly for the decay term) converges to

$$\lim_{\sigma\to 0}\int_{-\infty}^{\infty} \gamma_{\beta\sigma}(\omega)\hat{A}(\omega)[\cdot]\hat{A}(\omega)^{\dagger}d\omega = \sum_{\nu\in B(H)}\gamma_0(\nu)A_\nu[\cdot](A_\nu)^{\dagger}.$$

When we expand the left-hand side as a sum over $A_{\nu_1}[\cdot](A_{\nu_2})^{\dagger}$, the coefficients for each $\nu_1, \nu_2$

$$\lim_{\sigma\to 0}\int_{-\infty}^{\infty}\frac{\gamma_{\beta\sigma}(\omega)}{\sigma\sqrt{2\pi}}e^{-\frac{(\omega-\nu_1)^2}{4\sigma^2}}e^{-\frac{(\omega-\nu_2)^2}{4\sigma^2}}d\omega = \mathbb{1}(\nu_1 = \nu_2)\gamma_0(\nu_1),$$

approach the corresponding value in the Davies generator. By contrast, the coherent term vanishes, because the decay term commutes with the Hamiltonian in the $\sigma \to 0$ limit and $\tanh(0) = 0$ (see Lemma 1).

## Implementation

In this section, we give low-level quantum algorithmic implementations of our Lindbladian dynamics.

**A phase estimation for operators.** The most general form of the discrete operator Fourier transform is a subroutine similar to quantum phase estimation: it combines controlled Hamiltonian simulation with the quantum Fourier transform, as shown in Fig. 3a, acting on the time–frequency and system registers. To approximately implement the (continuous) operator Fourier transform, we need to discretize the time integral by introducing a time mesh $\bar{t} \in S_{t_0}$ and a corresponding frequency mesh $\bar{\omega} \in S_{\omega_0}$ for the QFT, each having $M$ points, resulting in the discretized operation

$$\underbrace{\sum_{\bar{t}\in S_{t_0}}f(\bar{t})|\bar{t}\rangle\otimes A \to \sum_{\bar{\omega}\in S_{\omega_0}}|\bar{\omega}\rangle\otimes\hat{A}(\bar{\omega})}_{\text{discrete operator Fourier transform}}$$

$$\text{where } \hat{A}(\bar{\omega}) := \frac{1}{\sqrt{M}}\sum_{\bar{t}\in S_{t_0}}e^{-i\bar{\omega}\bar{t}}f(\bar{t})e^{iHt}Ae^{-iHt}. \tag{25}$$

As given in Corollary C.2 of ref. 22, choosing

$$\omega_0 = 2\sigma\sqrt{\frac{2\pi}{M}}, \; t_0 = \frac{1}{2\sigma}\sqrt{\frac{2\pi}{M}},$$

$$S^{[M]} := \{-\lceil(M-1)/2\rceil, ..., -1, 0, 1, ..., \lfloor(M-1)/2\rfloor\}, \tag{26}$$

$$\text{and} \quad S_{\omega_0} := \omega_0 \cdot S^{[M]}, \quad S_{t_0} := t_0 \cdot S^{[M]}, \tag{27}$$

the above equation (25) recovers the advertised continuous operator Fourier transform (equation (13)) in the $M \to \infty$ limit. As $f$ is a smooth Gaussian function, we can achieve any finite precision $\epsilon$-approximation of the dissipative part of $\mathcal{L}$ with a moderately scaling dimension $M \sim \text{Poly}(\|H\|, \beta, 1/\epsilon, \sigma + 1/\sigma, |A|)$, which requires only $\log(M) = \tilde{\mathcal{O}}(1)$-many ancilla qubits, and no more than $\mathcal{O}(\sqrt{\log(1/\epsilon)}/\sigma)$ (controlled) Hamiltonian simulation time, because of the truncation of the Gaussian tail.

**A general-purpose Lindbladian simulation algorithm.** Another key algorithmic component is an improved black-box Lindbladian simulation subroutine. It achieves the sought nearly linear dependence on $t$ even for Lindbladians with high Kraus rank, as long as the Lindblad operators are given in a block-encoded form, which represents an improvement on previous Lindbladian simulation algorithms[51,57].

**Definition 1: block encoding for Lindblad operators.** (Definition I.2 of ref. 22) We say that a unitary $U$ is a block encoding for Lindblad operators $\{L_j : j \in J\}$, if

$$(\langle 0^b| \otimes I) \cdot U \cdot (|0^q\rangle \otimes I) = \sum_{j\in J}|j\rangle \otimes L_j \text{ for some } b \leq q \in \mathbb{N}.$$

In particular, discretized operator Fourier transforms (equation (25)) naturally give block encodings of this form, where $J$ refers to the discretized set of frequency labels $S_{\omega_0}$. Given a block encoding $U$ of the Lindblad operators as above, we can directly obtain a block encoding of the decay term $D$ by a single use of $U$ and $U^{\dagger}$ by standard

multiplication of block-encoded matrices[11]. As a consequence of the exponentially decaying tails in the integral representation of equation (20), by using an additional $\sim \beta$-time controlled Hamiltonian simulation and the linear combination of unitaries (LCU) technique[55], we can also obtain a sufficiently good approximate block encoding of the coherent term $C$.

We begin with a brief analysis of the first-order simulation circuit shown in Fig. 3b. Assuming the system register is in the pure state $|\psi\rangle$, the first three gates act as follows (for simplicity, we drop the superscripts $A^a \leftarrow A$):

$$
\begin{aligned}
|0\rangle|0^q\rangle|\psi\rangle \;&\overset{(1)}{\to}\; |0\rangle \otimes \sum_{\overline{\omega} \in S_{\omega_0}} |\overline{\omega}\rangle \otimes \hat{A}(\overline{\omega})|\psi\rangle \\[4pt]
&\overset{(2)}{\to}\; \sum_{\overline{\omega} \in S_{\omega_0}} \left( \underset{1 - \frac{\delta\gamma(\overline{\omega})}{2} + \mathcal{O}(\delta^2)}{\sqrt{1 - \delta\gamma(\overline{\omega})}\,|0\rangle + \sqrt{\delta\gamma(\overline{\omega})}\,|1\rangle} \right) |\overline{\omega}\rangle \\[4pt]
&\qquad \hat{A}(\overline{\omega})|\psi\rangle \\[4pt]
&\overset{(3)}{\to}\; |0\rangle|0^q\rangle\left( I - \frac{\delta}{2}D \right)|\psi\rangle + |1\rangle \sum_{\overline{\omega} \in S_{\omega_0}} \sqrt{\delta\gamma(\overline{\omega})}\,|\overline{\omega}\rangle \\[4pt]
&\qquad \hat{A}(\overline{\omega})|\psi\rangle - \frac{\delta}{2}|0\rangle \otimes |0^q \perp\rangle + \mathcal{O}(\delta^2),
\end{aligned}
\tag{28}
$$

where $|0^q \perp\rangle$ is a quantum state such that $\| |0^q\perp\rangle \| \leq 1$ and $(\langle 0^q| \otimes I)\cdot|0^q\perp\rangle = 0$, see Theorem III.1 of ref. 22, for details.

Let $|\psi'\rangle$ be the resulting state in equation (28), and let us denote the dissipative part $\mathcal{L}' := \mathcal{L}^a + i[C^a, \cdot]$. Tracing out the first $q+1$ qubits, we get that $|\psi'\rangle$ is $\mathcal{O}(\delta^2)$-close to the desired state, ignoring the coherent term. We now show that

$$
\left\| (\mathcal{I} + \delta\mathcal{L}')[|\psi\rangle\langle\psi|] - \mathrm{Tr}_{q+1}[|\psi'\rangle\langle\psi'|] \right\|_1 = \mathcal{O}(\delta^2)
\tag{29}
$$

by observing that

$$
\begin{aligned}
\mathrm{Tr}_{q+1}[|\psi'\rangle\langle\psi'|] &= \mathrm{Tr}_q[(\langle 0| \otimes I)\cdot|\psi'\rangle\langle\psi'|\cdot(|0\rangle \otimes I)] \\[4pt]
&\quad + \mathrm{Tr}_q[(\langle 1| \otimes I)\cdot|\psi'\rangle\langle\psi'|\cdot(|1\rangle \otimes I)] \\[4pt]
&= \left( I - \frac{\delta}{2}D \right)|\psi\rangle\langle\psi|\left( I - \frac{\delta}{2}D \right) \\[4pt]
&\quad + \delta \sum_{\overline{\omega} \in S_{\omega_0}} \gamma(\overline{\omega})\hat{A}(\overline{\omega})|\psi\rangle\langle\psi|\hat{A}(\overline{\omega})^\dagger + \mathcal{O}(\delta^2) \\[4pt]
&= (\mathcal{I} + \delta\mathcal{L}')[|\psi\rangle\langle\psi|] + \mathcal{O}(\delta^2).
\end{aligned}
$$

Convexity and the triangle inequality imply that equation (29) also holds for mixed input states. By including the $\delta$-time Hamiltonian simulation for $C^a$, we get an $\mathcal{O}(\delta^2)$ approximation of $\delta$-time evolution by $\mathcal{L}^a$. Once again, owing to linearity and the triangle inequality, this also implies that performing the circuit Fig. 3b for a uniformly random jump $A^a$ we get a $\delta$-time evolution by $\frac{1}{|A|}\sum_{a \in A}\mathcal{L}_a$ up to $\mathcal{O}(\delta^2)$ error in trace distance, see Corollary III.1 of ref. 22. Repeating the entire argument after replacing the jump operators $A^a$ by $A^a \otimes I$, we can see that actually the above results in a $\mathcal{O}(\delta^2)$-precise implementation of the quantum channel $\exp\left(\frac{\delta}{|A|}\sum_{a \in A}\mathcal{L}_a\right)$ in the completely bounded 1-1 superoperator norm, that is, the diamond-norm.

Choosing $\delta = \Theta\left(\frac{\epsilon}{t}\right)$ ensures that the error in a single time step is bounded by $\mathcal{O}\left(\frac{\epsilon^2}{t^2}\right)$, and repeating the process $\Theta\left(\frac{t^2}{\epsilon}\right)$-times induces an error that is bounded by $\epsilon$ for the entire time-$t$ evolution. The complexity is then $\Theta\left(\frac{t^2}{\epsilon}\right)$ times the cost of implementing the circuit in Fig. 3b.

Building on the compression techniques in ref. 57, we can bootstrap the first-order weak-measurement circuit of Fig. 3b by observing that for very small time steps, the circuit is very close to identity. Exploiting

this by effectively only using the circuit Fig. 3b in parts of the trajectories that are nontrivial, we can achieve almost linear scaling in $t$ and polylogarithmic scaling in the desired diamond-norm accuracy of the simulation. However, we need to apply some modifications, as it turns out the original compressed measurement scheme described in ref. 57 does not work as intended. Thus, we use a variant of the analogous measurement scheme described in ref. 58 (see Appendix F of ref. 22, for more details). This amounts to our ultimate near-linear-time simulation result.

**Theorem 2: almost linear-time Lindbladian simulation.** (Theorem III.2 of ref. 22) Suppose $U$ is a unitary block encoding of the Lindblad operators of $\mathcal{L}$ as in Definition 1, and $V$ is a block encoding of the coherent term $C$. Let $t \geq 1$ and $\epsilon \leq 1/2$, then we can simulate the map $e^{\mathcal{L}t}$ to error $\epsilon$ in diamond norm using

$$
\mathcal{O}((q + \log(t/\epsilon))\log(t/\epsilon)) \quad \text{(resettable) ancilla qubits,}
\tag{30}
$$

$$
\mathcal{O}\left(t\frac{\log(t/\epsilon)}{\log\log(t/\epsilon)}\right) \quad \text{(controlled) uses of } U, V, U^\dagger \text{ and } V^\dagger,
\tag{31}
$$

$$
\text{and} \quad \mathcal{O}(t(q+1)\mathrm{polylog}(t/\epsilon)) \quad \text{other two-qubit gates,}
\tag{32}
$$

where $q$ is the number of ancilla qubits used for the block encodings.

To place the above general result in context, we give some simple bounds on the resources required for implementing our Lindbladian. For instance, if the jump operators are $K$ different Pauli strings, then $q = \log_2(K) + 1$ ancilla qubits suffice for block encoding them. The overall gate complexity should be dominated by the controlled Hamiltonian simulation subroutine. Thus, we focus on estimating the required Hamiltonian simulation time per call to $U$ and $V$ (assuming an operator Fourier transform width $\sigma = \frac{1}{\beta}$). Under the normalization $\|\sum_{a \in A} A^{a\dagger}A^a\| \leq 1$, the Lindblad operators from our construction (equation (7)) can be $\epsilon$-accurately block-encoded by discretized operator Fourier transform using $\mathcal{O}(\beta\sqrt{\log(1/\epsilon)})$ (controlled) Hamiltonian simulation time by truncating the Gaussian integrand in equation (6). Meanwhile, as implied by Corollary III.2 of ref. 33, a slightly subnormalized coherent term $C/\alpha$ for $\alpha = \mathcal{O}\left(\log\left(\frac{\beta\|H\|}{\epsilon}\right)\right)$ can be $\epsilon$-accurately block-encoded by LCU by using a mere $\mathcal{O}(\beta\log(1/\epsilon))$ (controlled) Hamiltonian simulation time. The extra subnormalization factor $\alpha$ can be absorbed into the number of uses of the block encoding $V$ using state-of-the-art block-encoded Hamiltonian simulation techniques[4,11].

To approximate $e^{\mathcal{L}t}$ to $\epsilon$-diamond distance, we can control the accumulation of errors by setting an increased accuracy goal of $\mathrm{poly}(\epsilon/(t\beta\|H\|))$ for the approximate block encodings $U, V$. This results in an overall $\mathcal{O}(\beta\sqrt{\log(t\beta\|H\|/\epsilon)})$ and $\mathcal{O}(\beta\log^2(t\beta\|H\|/\epsilon))$ (controlled) Hamiltonian simulation time overhead for implementing sufficiently accurate block encodings $U$ and $V$, respectively.

## Comparison with physically derived master equations

Our synthetic Lindbladian is mainly presented as an algorithmic construction. In this section, we compare it with physically motivated master equations derived from first-principles open-system calculations. Overall, we believe that our Lindbladian can serve as a self-contained toy model of thermalization. Nevertheless, for quantitative modelling of particular system–bath interactions, the role of strict detailed balance is less clear, and it may be preferable to use a physically derived master equation.

The textbook setup in open-system thermalization[24,59] considers a system governed by a Hamiltonian $H_S$ that couples weakly to a large thermal bath with Hamiltonian $H_B$, which are together governed by the total Hamiltonian $H_{\mathrm{tot}} = H_S \otimes I_B + I_S \otimes H_B + \lambda \sum_{a \in A} A^a \otimes B^a$. The Hermitian operators $\{A^a : a \in A\}$ act on the system (mirroring our jump

operators, hence the same notation) and $\{B^a : a \in A\}$ acts on the bath, whereas $\lambda$ represents the coupling strength. Tracing out the bath, we can obtain an effective master equation governing the system dynamics under the assumptions that the thermal bath is Markovian and the coupling $\lambda$ is sufficiently weak.

The aforementioned Davies generator was originally derived in the weak-coupling limit $\lambda \to 0$ (relative to all other energy scales). In this limit, the rotating-wave approximation (or the secular approximation) removes the cross terms $A_{\nu_1}[\,\cdot\,]A_{\nu_2}^\dagger$; this perfect isolation of Bohr frequencies causes the Davies generator to satisfy detailed balance, but at the same time, makes it implausible for many-body systems with exponentially small level spacings. To focus on issues related to detailed balance, here we studied only the simplest form of the Davies generator. In principle, different jumps $A^a$ may have different transition weights, as they correspond to the Fourier transform of the bath correlation function $\langle e^{iH_\beta t} B^a e^{-iH_\beta t} B^a \rangle$, but for algorithmic purposes, it is natural to use universal transition weights. Also, we have studied only the dissipative part of the Davies generator (equation (5)). However, the full Davies generator also includes a Lamb-shift term that somewhat resembles our coherent term, but depends on additional details of the bath correlation functions. As the Lamb-shift term commutes with the Hamiltonian, adding this term keeps the Gibbs state stationary.

Recently, there have been several attempts to derive more realistic master equations for many-body systems by avoiding the $\lambda \to 0$ limit needed for the rotating-wave approximation. This essentially translates to introducing Lindblad operators with a finite energy uncertainty, or, equivalently, in the time domain, integrals over Heisenberg-evolved jump operators weighted by a finite-width window function. In particular, the coarse-grained master equation[59] takes a very similar form as ours (equation (7)), but its operator Fourier transform features a uniform integral $\frac{1}{\sqrt{2\pi T}}\int_{-\frac{T}{2}}^{\frac{T}{2}} e^{iHt} A^a e^{-iHt} e^{-i\omega t}\,\mathrm{d}t$, and its coherent term also resembles ours (equation (19)) but does not seem to strictly enforce detailed balance. The universal Lindblad equation[60] combines (square root of) the transition weights into the operator Fourier transforms and is closer to subsequent constructions[61,62]. Very recently[63] derived a master equation with exact KMS-detailed balance using slightly different operator Fourier transform weights compared with ours. All the above recent master equations feature some finite energy uncertainty, derived from various system–bath parameters, that parallels our tunable Gaussian width $\sigma$.

## Data availability

The data that support the findings of this study are available from the corresponding author upon reasonable request.

## Code availability

The code for numerical plots is available upon reasonable request.

51. Rall, P., Wang, C. & Wocjan, P. Thermal state preparation via rounding promises. *Quantum* **7**, 1132 (2023).
52. Ramkumar, A. & Soleimanifar, M. Mixing time of quantum Gibbs sampling for random sparse Hamiltonians. Preprint at arxiv.org/abs/2411.04454 (2024).
53. Guo, J., Hart, O., Chen, C.-F., Friedman, A. J. & Lucas, A. Designing open quantum systems with known steady states: Davies generators and beyond. *Quantum* **9**, 1612 (2025).
54. Amorim, É. & Carlen, E. A. Complete positivity and self-adjointness. *Linear Algebra Appl.* **611**, 389–439 (2021).
55. Childs, A. M. & Wiebe, N. Hamiltonian simulation using linear combinations of unitary operations. *Quantum Inf. Comput.* **12**, 901–924 (2012).
56. Wolf, M. M. *Quantum Channels & Operations: Guided Tour* https://mediatum.ub.tum.de/download/1701036/1701036.pdf (2012).
57. Cleve, R. & Wang, C. Efficient quantum algorithms for simulating Lindblad evolution. In *Proc. 44th International Colloquium on Automata, Languages, and Programming (ICALP 2017)* (eds Chatzigiannakis, I., Indyk, P., Kuhn, F. & Muscholl, A.) 17–11714 (2017).
58. Berry, D. W., Cleve, R. & Gharibian, S. Gate-efficient discrete simulations of continuous-time quantum query algorithms. *Quantum Inf. Comput.* **14**, 1–30 (2014).
59. Mozgunov, E. & Lidar, D. Completely positive master equation for arbitrary driving and small level spacing. *Quantum* **4**, 227 (2020).
60. Nathan, F. & Rudner, M. S. Universal Lindblad equation for open quantum systems. *Phys. Rev. B* **102**, 115109 (2020).
61. Ding, Z., Li, B. & Lin, L. Efficient quantum Gibbs samplers with Kubo–Martin–Schwinger detailed balance condition. *Commun. Math. Phys.* **406**, 67 (2025).
62. Gilyén, A., Chen, C.-F., Doriguello, J. F. & Kastoryano, M. J. Quantum generalizations of Glauber and Metropolis dynamics. Preprint at arxiv.org/abs/2405.20322 (2024).
63. Scandi, M. & Alhambra, Á. M. Thermalization in open many-body systems and KMS detailed balance. Preprint at arxiv.org/abs/2505.20064 (2025).
64. Moussa, J. E. Low-depth quantum Metropolis algorithm. Preprint at arxiv.org/abs/1903.01451 (2019).
65. Moussa, J. E. Quantum Metropolis-Hastings algorithm. Preprint at arxiv.org/abs/2503.14970 (2025).

**Acknowledgements** We thank J. Moussa for raising the question of whether exact detailed balance is possible and pointing us to his related work[64,65]. We also thank A. J. Friedman, J. Guo, O. Hart and A. Lucas for collaborations on related inspiring topics[53]. C.F.C. was supported by the Caltech Eddlemen Fellowship and the AWS Center for Quantum Computing internship, and later by a Simons-CIQC postdoctoral fellowship through NSF QLCI Grant No. 2016245. A.G. is supported by the AWS Center for Quantum Computing. M.J.K. acknowledges support from the Novo Nordisk Foundation (grant no. NNF23OC0083524). We also thank the Simons Institute for hosting C.F.C. and A.G.

**Author contributions** The thread of ideas for this study was initiated by discussions between M.J.K. and F.B. in 2016 and was later brought to fruition by C.F.C. and A.G. The main calculations and proofs were developed by C.F.C. and A.G., with C.F.C. focusing on the analytic properties and A.G. on the quantum algorithmic constructions. A.G. and M.J.K. collaborated on the numerical aspects. All authors contributed to the conceptual discussions and manuscript writing.

**Competing interests** A.G. acknowledges funding from the AWS Center for Quantum Computing. Other authors do not have any competing interests.

**Additional information**
**Correspondence and requests for materials** should be addressed to Chi-Fang Chen.
