## [Peer Review file · Nature]

Efficient Quantum Thermal Simulation

Corresponding Author: Dr Chi-Fang Chen

Version 1:

Reviewer comments:

Referee #1

(Remarks to the Author)

A: Summary

The authors introduce a novel quantum Gibbs sampler, i.e. a quantum algorithm to prepare quantum Gibbs states, that satisfies various desiderata like:

1. it satisfies exact detailed balance (i.e. it is self-adjoint w.r.t to an appropriate scalar product). This is a very convenient mathematical property of Gibbs samplers and ensures that the quantum Gibbs states of interest is the fixed point of the evolution.
2. It has quasi-local updates, meaning that the generator of the evolution can be decomposed into quasi-local terms.
3. It can be implemented efficiently in a quantum computer whenever we can efficiently simulate the underlying Hamiltonian.

To the best of my knowledge, this is the first example of quantum Gibbs samplers that have all of these properties, although several works, including some by the authors of this manuscript, have since generalized the construction.

One can name at least two reasons why this work is extremely interesting: first, Markov Chain Monte Carlo methods are one of the most useful meta algorithms in classical computing, with a wide range of applications, and the algorithm put forth here is arguably a quantum version of that and has the potential to mirror its success. Second, preparing states of quantum many-body systems is arguably one of the main envisioned applications of quantum computers, and it is likely that the algorithm discussed here will be a powerful tool to prepare such states.

The main technical insight of the authors is to consider an appropriately weighted operator Fourier transform of jump operators that ensures a Lindbladian with quasi-local jumps. In addition, to simultaneously ensure detailed-balance, they add a carefully engineered coherent term to the evolution, which is absent in the classical case.

The potential of this generator has been since confirmed in various studies that show that it allows for the efficient preparation of high temperature quantum Gibbs states of spin systems and certain fermionic systems, here even reaching the ground state.

Besides laying down the theoretical framework surrounding the Lindbladians, the authors perform numerical experiments for small system sizes and $1D$ systems and attempt to give a physical intuition about them.

B: Originality and significance, C, Data & methodology and H: Clarity and context

The results of this paper are of central importance to quantum computing and simulation of quantum many-body systems. It unlocked a novel meta algorithm to prepare quantum many-body states which has since been shown to indeed be able to prepare various states, giving evidence it will be of importance also in practice. In addition, it was also an important technical tool to solve other central problems in the theory of quantum many-body states, such as the recent result that Gibbs states are locally Markovian by one of the authors. Thus, I am confident it will continue to be of both conceptual and practical value to the study of quantum many-body systems.

Furthermore, I am confident that the results presented in the paper are correct.

That being said, I believe that the authors could have done a better job at presenting their construction to a general audience. It might be difficult to grasp the motivation behind the construction and certain key aspects of the construction are only commented on. For instance, the quasi-locality of jumps: the authors just say that one can use Lieb-Robinson bounds to show it, but do not formulate precisely what it is supposed to mean or how to show it. They also repeatedly say that for certain choices of parameters they recover well-known algorithms like Glauber dynamics, but do not do the whole argument. As Glauber or Metropolis dynamics should be more familiar territory for a larger subset of the audience, I think this could be of pedagogical value.

It might be difficult for me to judge this aspect of the manuscript objectively as I am an expert on the field, but my impression is that a reader with a more general physics (or maybe even quantum computing background) might find it difficult to follow the construction and I am afraid that its importance might get lost in the technicalities.

In addition, I am a bit skeptical that the numerics adds much value to the manuscript. This is because the authors only study systems comprised of at most 8 qubits, which are system sizes too small to draw strong conclusions. Only focusing on one system size also seems restrictive to me.

The same can be said about the section on Physical motivations. At its current state it is quite hand-wavy and I do not believe it warrants the phrase "a direct connection to the open system dynamics of weak coupling to a Markovian thermal bath." To the best of my knowledge, it is still unclear how to make this construction arise naturally from in physical systems and this section of the paper did not convince me further.

D,E

Do not apply, as this is a theoretical work.

F: suggested improvements

Here is a list of typos and detailed comments:

Fixed-Point Uniqueness (p. 6)

- **"Similarly to the classical case"** should replace **"Similatly to the classical case"**
- **"trace-preservation of Lindbladians"** should replace **"trace-reservation of Lindbladians"**

Figure 3 Label

- **"Quantum Operator Fourier Transform"** should replace **"Quantum Operator Foruier Transform"**

References

- **Missing publication years ("???)"**
Several references show incomplete publication information (e.g., "Oxford University Press, ??? (2007)", "Academic Press, ??? (1965)"). Please add the missing years.
- Some technical terms are not properly introduced, such as discriminant in the figure or diamond norm.
- I do not like the use of $\$c\$$ for the block encoding. I would reserve this letter for constants.
- Figure 4: it would be better to have log-plot to get a more quantitative understanding of the gap. You could also mention that the gap is known classically for the case $\lambda=0$ to guide the reader as to what to expect.
- It might be of pedagogical value to present a simple, small example to showcase the technique.

Conclusion

I believe that this is a very important contribution at the intersection of quantum computing and quantum many-body systems. I am convinced it will play an important role both conceptually and practically for these areas in the years to come.

However, for it to be published at a journal with a wide readership and high standards like Nature, it is imperative that the presentation is improved to a general audience. In addition, the numerics section needs to be improved to be more informative and the "Physical motivations" section significantly strengthened. Alternatively, the authors could admit a shortcoming of this work say that the physical mechanisms behind this construction still need to be further investigated.

G: references

The references seem appropriate.

Referee #2

(Remarks to the Author)

In this manuscript, the authors propose an efficient quantum algorithm that generalizes classical Markov chain Monte Carlo (MCMC) algorithms. While classical MCMC algorithms promise to sample from thermal distributions of classical Hamiltonians, the authors' algorithm generates samples from the thermal state of a quantum Hamiltonian. Classical MCMC algorithms have had a tremendous impact in (and beyond) physics over the last 75 years and I agree with the authors that a quantum analogue may be similarly impactful to the study of quantum systems.

There is a variety of prior work that attempts to address the preparation of quantum Gibbs states, but the proposal outlined in this paper presents substantial advantages compared to these prior attempts. Some of these algorithms work quite generally, but they scale exponentially with the inverse temperature β . Other algorithms have attempted to emulate physical cooling processes in various ways, but, as the authors point out, prior attempts to derive efficient and assumption-free quantum analogues of MCMC have failed. It has been an open question whether or not such a quantum algorithm exists, and this paper (and related work by an overlapping set of authors) resolves it affirmatively.

Like classical MCMC algorithms, this quantum algorithm is actually a family of algorithms that can be tailored to incorporate physical understanding and intuition. As the authors explain, their algorithm offers the user a wide range of freedom in choosing the jump operators and the transition weights. I appreciated the inclusion of Figure 4, which shows some particular toy examples where these degrees of freedom make a large impact on the mixing time (and therefore the overall feasibility of using the algorithm).

The quality of the presentation was generally excellent, and I think the authors did an admirable job in presenting the highly technical content in a way that will be accessible to a broader audience. Or, at least as accessible as one can hope. If accepted into Nature, the number and length of the equations in this paper will be significantly higher than the average manuscript, but I think that this may be unavoidable given the topic. Still, if there are any additional aspects of the technical presentation in the main text that the authors could shift to the main text, I think that it might benefit the typical reader.

Based on the significance of the results and the broad applicability of classical MCMC methods, I recommend that the paper be accepted, although I would understand if the editorial decision was to instead recommend transfer to Nature Physics.

I will include a handful of minor suggestions below:

I think the sentence at the beginning of Section 1.1 would read better if the "Remarkably," were omitted. The results really stand on their own.

The fact that the Hamiltonian simulation time per unit time scales as β is obviously much better than something that scales exponentially with β , but I found it interesting that the cost of a "step" scales with the inverse temperature. Do classical algorithms have a similar feature if we take into account, say, the rejection probability? Maybe a brief comment on this in the text could help provide intuition for readers who are familiar with classical MCMC algorithms.

In the text of Figure 3, "Foruier" should be "Fourier."

A few of the references have missing or duplicated links, or have some text that reads "???"

Referee #3

(Remarks to the Author)

Markov Chain Monte Carlo (MCMC) methods have become indispensable tools in classical many-body simulation. Metropolis sampling and Glauber dynamics, in particular, have been successful in sampling from thermal distributions for many physically-motivated classical systems and, as a result, have proven to be one of the most widely-used class of algorithms throughout science. A major goal of quantum algorithms has been to replicate this success by developing a quantum analog to MCMC methods for preparing Gibbs states of quantum systems on a quantum computer. This paper presents the first algorithm to achieve this goal. The algorithm presented here can be efficiently implemented on a quantum computer and achieves the detailed balance condition, guaranteeing that the Gibbs state is a stationary state of the process. With the additional assumption that the chosen jump operators are sufficiently mixing, the Gibbs state is then the unique fixed point.

This paper gives an analytical description of the dissipative process in the form of a Lindbladian, the quantum analog of continuous Markov chains and then shows how the Lindbladian can be efficiently simulated on a quantum computer with a running time that depends linearly on the length of the simulation. Showing that the process can efficiently prepare Gibbs states still depends on establishing a bound on the mixing time of the process. Subsequent work has already shown that the Lindbladian-based algorithm presented here converges rapidly for Hamiltonians on a finite-dimensional lattice at sufficiently high temperature, giving the first efficient algorithm to compute Gibbs states in these cases.

Simulating quantum systems is expected to be one of the primary applications for quantum computers of the future. There has been significant progress over the years in simulating the dynamics of quantum systems. However, algorithms for simulating the equilibrium properties of quantum many-body systems have been much more challenging. Prior to this work,

there has not been a candidate for a widely applicable algorithm for preparing physically-motivated quantum states, as the Metropolis algorithm has been for classical systems. The algorithm presented in this paper is a strong candidate to be one of the most important algorithms for quantum simulation for future quantum computers.

A small section of the paper presents numerical results showing that the methods presented will converge efficiently with specific Hamiltonians and temperatures, suggesting that the method will be a useful heuristic in particular scenarios. The bulk of the paper is based on mathematical proof that the Lindbladian has the detailed balance condition and can be efficiently simulated as a function of the error and simulation length.

For the most part the paper is well-written. I have organized my comments on the draft to general comments to improve clarity, followed by a list of small typos:

Caption of Figure 1, in the sentence starting with "However,..": It's not clear what explicit descriptions of energy states and cloning quantum states have to do with the rejection step. It seems that the difficulty in rejecting has to do with the fact that a measurement is involved in determining whether to accept or reject. Then rejecting requires backing up to the previous state and there is no clear way to do this without cloning the starting state or having an explicit description of the state.

Line 117: I'm not sure what the bath correlation function is or how it determines the function γ . Can you say a little more?

Line 120: I was initially confused by the sentence that starts with "More generally..." I think you are still assuming the diagonal input states and jumps that preserve the diagonal states mentioned in the previous sentence. It would be helpful to clarify this point.

Line 149: This is the first mention of the shifted Metropolis transition weights and it wasn't clear where it came from. A hint to the reader that this will be derived later would be helpful: "We derive in Section X that, remarkably, exact detailed balance can be achieved with Gaussian energy uncertainty when shifted Metropolis transition weights are used."

Line 191: Even after reading the rest of the paper, it's still a little mysterious to me why $O(\beta)$ Hamiltonian simulation time suffices to ensure detailed balance. Can you say more?

Line 275: Why does detailed balance require $\sigma \geq 1/\beta$? I would suggest a forward pointer to where this is derived or some intuition here. (Or both)

Line 284: "As we derive in Section X, the shift in energy difference..."

Line 287: What's so convenient about $\sigma \geq 1/\beta$? I guess this is the condition that guarantees detailed balance but, again, some intuition or a forward pointer to a more careful explanation would be helpful.

Page 15: Is the derivation of detailed balance supposed to depend on the assumption that $\sigma \geq 1/\beta$ as previously stated? I would think it would have more to do with the error bounds in discretizing the infinite integral in the implementation.

Line 334: Is the reason that the closed form implies algorithmic efficiency that the exponential decay in t allows you to truncate the infinite integral? And does the quasi-locality require a Lieb-Robinson bound? If so, this should probably be explicit.

Line 358: Does the relationship between N and ϵ implicitly use the fact that the width σ is chosen to be $1/\beta$?

Line 374: You should probably remind the reader that Fig 3A is a simplified, less-efficient Lindbladian simulation algorithm and that you will discuss the more efficient one later.

Line 400: Theorem 2 doesn't give any indication of the dependence of the running time on ϵ . There is at least a linear factor in $1/\epsilon$, right? Also, the variable c used in the theorem is the same c from the size of the second register first used on page 18, but it seems to be a free-floating variable, and there's no indication here about what constraints govern the choice of the parameter c .

I think I would have preferred a final theorem statement that gives the running time for the specific Lindbladian from (9) as a function of running time t and error ϵ , using only a black-box algorithm to simulate the dynamics of the Hamiltonian e^{iHt} for one unit of time. At the end of the day, that's what you want to do, right?

Typos:

Page 5, line 112: "the Davies generator" and "a certain Fourier transform"

Line 282: Do you mean "constructed"?

Line 339: Similarly

Line 363: algorithms

374: in Figure 3A

Dear Reviewers,

We would like to thank you for your time and effort in providing valuable assessments of our manuscript. We have incorporated the feedback, and noteworthy revisions are highlighted in blue. We have also made minor changes throughout the entire manuscript. Your comments have been very useful in enhancing the readability of our manuscript, and we believe it is now more accessible to a broad audience.

First, we summarize the main revisions that address the general points raised by the referees.

- The referees are concerned that the presentation may still be too technical for a general audience. In response, we have reconciled all large blocks of equations throughout the paper, retaining only the essence of the arguments while deferring the interested readers to details in the cited technical drafts.
 - We have streamlined the proof of detailed balance for our construction by breaking it into individual steps, starting with demonstrating detailed balance for Davies' generator and exposing useful symmetries of the operator Fourier transform (from line 414).
 - We have significantly reduced the equations for the Lindbladian simulation algorithm (from line 565).
- Regarding the concern that some claims are not self-contained or explained, we implement changes according to the referees' suggestions as follows:
 - (line 515) We included a single-qubit explicit example to compare Davies' generator and our construction. We have included only the transition part; however, if the referees would also like to see the coherent term and the length permits, we would be happy to include it.
 - (line 526) We explain how the algorithm recovers Davies' generator in suitable limits.
 - (line 491) We included a minimal discussion on Lieb-Robinson bounds. Since the locality estimates are routine yet somewhat more involved to state precisely (and often depend on the particular application), we refer to the literature for a full exposition.
 - (line 199) We added a dedicated paragraph after Theorem 1 addressing "why $O(\beta)$ Hamiltonian simulation time suffices to ensure detailed balance", by emphasizing that high-precision metrology is not needed for detailed balance, and commenting that this β scaling is inherent in the noncommutativity of the Hamiltonian terms. We added a deeper discussion about it in the Methods under the "Exact detailed balance with finite uncertainty" paragraph (line 450).
- The referees were concerned with the connection between our construction and the physical microscopic mechanism of thermalization. (line 612) We acknowledge that this connection is intended to be qualitative and have reworded the paper accordingly, using

phrases such as “toy model for thermalization” and revising the physical motivation paragraph to clarify, including a brief comparison to physically derived master equations.

In the remainder of this letter, we repeat your concrete suggestions and address them one by one.

Ref #1

That being said, I believe that the authors could have done a better job of presenting their construction to a general audience. It might be difficult to grasp the motivation behind the construction, and certain key aspects of the construction are only commented on. For instance, the quasi-locality of jumps: the authors just say that one can use Lieb-Robinson bounds to show it, but do not formulate precisely what it is supposed to mean or how to show it. They also repeatedly say that for certain choices of parameters they recover well-known algorithms like Glauber dynamics, but do not do the whole argument. As Glauber or Metropolis dynamics should be more familiar territory for a larger subset of the audience, I think this could be of pedagogical value.

- As summarized, we have revised accordingly to better present and explain the construction (line 432). For the last point on Glauber or Metropolis, we have included how the Davies generator gets recovered in a certain limit of our Lindbladian (line 526), and commented that the Davies generator further reduces to Glauber dynamics.

It might be difficult for me to judge this aspect of the manuscript objectively as I am an expert on the field, but my impression is that a reader with a more general physics (or maybe even quantum computing background) might find it difficult to follow the construction and I am afraid that its importance might get lost in the technicalities.

- As summarized, we paid attention to make the manuscript as self-contained as possible, gave a step-by-step derivation of detailed balance, while we have also streamlined all large blocks of equations and added explanatory texts.

In addition, I am a bit skeptical that the numerics adds much value to the manuscript. This is because the authors only study systems comprised of at most 8 qubits, which are system sizes too small to draw strong conclusions. Only focusing on one system size also seems restrictive to me.

- (line 219) We agree that looking at an 8-qubit system does not say much about the thermodynamic limit. However, the main role of the numerics is (1) to serve as a sanity check for our construction on the simplest models and (2) to illustrate the important role of the choice of the jump operators. Also, we note that larger numerics are prohibitive, as the reliable direct diagonalization methods we use would already require more than 1 TB of memory, since we need to analyze (dense) superoperators. Due to parity issues, it mostly makes sense to compare models with the same qubit number parity. We have also plots for $N=4$ and 6 , but they look qualitatively similar, and do not seem to add much

value to the reader, so we did not include them. Also to mitigate finite-size effects we use periodic boundary conditions.

The same can be said about the section on Physical motivations. At its current state it is quite hand-wavy and I do not believe it warrants the phrase "a direct connection to the open system dynamics of weak coupling to a Markovian thermal bath." To the best of my knowledge, it is still unclear how to make this construction arise naturally from in physical systems and this section of the paper did not convince me further.

- (line 612) As summarized, we have compressed the physical motivation section and clarify that we only claim qualitative resemblance to physically derived Lindbladians. We have also reworded through the draft in terms of "toy model of thermalization."

Figure 4: it would be better to have log-plot to get a more quantitative understanding of the gap. You could also mention that the gap is known classically for the case $\lambda=0$ to guide the reader as to what to expect.

- (Figure 4) We have added insets of log-plots, and also mention the $\lambda=0$ case in the caption.

Ref #2

However, for it to be published at a journal with a wide readership and high standards like Nature, it is imperative that the presentation is improved to a general audience. In addition, the numerics section needs to be improved to be more informative and the "Physical motivations" section significantly strengthened. Alternatively, the authors could admit a shortcoming of this work say that the physical mechanisms behind this construction still need to be further investigated.

- As summarized, we have attempted to make the paper more self-contained, and streamlined the presentation to reduce the volume of equations. (line 612) The physical motivation section is compressed, and we do not claim a quantitative connection to physical thermalization. We have also reworded throughout the draft in terms of "toy model of thermalization." We also added an inset of log plots to the figure summarizing the numerical results (Figure 4).

The quality of the presentation was generally excellent, and I think the authors did an admirable job in presenting the highly technical content in a way that will be accessible to a broader audience. Or, at least as accessible as one can hope. If accepted into Nature, the number and length of the equations in this paper will be significantly higher than the average manuscript, but I think that this may be unavoidable given the topic. Still, if there are any additional aspects of the technical presentation in the main text that the authors could shift to the main text, I think that it might benefit the typical reader.

- As an attempt to find common ground across referees' suggestions on the level of technicality, we have included only the minimal level of equations to fully define the construction in the main text. For the curious readers, we have also provided a step-by-step, streamlined derivation of the main results in the Methods section (from line 414).

The fact that the Hamiltonian simulation time per unit time scales as β is obviously much better than something that scales exponentially with β , but I found it interesting that the cost of a “step” scales with the inverse temperature. Do classical algorithms have a similar feature if we take into account, say, the rejection probability? Maybe a brief comment on this in the text could help provide intuition for readers who are familiar with classical MCMC algorithms.

- As summarized, we have clarified in the paragraph after Theorem 1 that β scaling is inherently due to noncommutativity; (line 210) we have clarified that commuting and classical Gibbs samplers can be implemented without a β dependence, and included a deeper explanation under the “Exact detailed balance with finite uncertainty” paragraph in Methods.

Ref# 3

Caption of Figure 1, in the sentence starting with “However,..”: It’s not clear what explicit descriptions of energy states and cloning quantum states have to do with the rejection step. It seems that the difficulty in rejecting has to do with the fact that a measurement is involved in determining whether to accept or reject. Then rejecting requires backing up to the previous state and there is no clear way to do this without cloning the starting state or having an explicit description of the state.

- We edited the caption to emphasize that the difficulty lies in “backing up” an unknown quantum state.

Line 117: I’m not sure what the bath correlation function is or how it determines the function γ . Can you say a little more?

- We have added a little more explanation in the “Comparison with physically derived master equations” part of the Methods.

Line 120: I was initially confused by the sentence that starts with “More generally...” I think you are still assuming the diagonal input states and jumps that preserve the diagonal states mentioned in the previous sentence. It would be helpful to clarify this point

- (Line 120) We have reworded the sentences to clarify that “the diagonal input states and jumps” are meant for both Glauber and Metropolis.

Line 149: This is the first mention of the shifted Metropolis transition weights and it wasn’t clear where it came from. A hint to the reader that this will be derived later would be helpful: “We derive in Section X that, remarkably, exact detailed balance can be achieved with Gaussian energy uncertainty when shifted Metropolis transition weights are used.”

- (line 153) We have added a pointer to the Methods where this is derived step by step.

Line 191: Even after reading the rest of the paper, it's still a little mysterious to me why $O(\beta)$ Hamiltonian simulation time suffices to ensure detailed balance. Can you say more?

- (line 199) As we summarized, we have added a paragraph after Theorem 1 to clarify that detailed balance is more like a symmetry and does not require high-precision metrology.

Line 275: Why does detailed balance require $\sigma \geq 1/\beta$? I would suggest a forward pointer to where this is derived or some intuition here. (Or both)

Line 284: "As we derive in Section X, the shift in energy difference..."

Line 287: What's so convenient about $\sigma \geq 1/\beta$? I guess this is the condition that guarantees detailed balance but, again, some intuition or a forward pointer to a more careful explanation would be helpful.

Page 15: Is the derivation of detailed balance supposed to depend on the assumption that $\sigma \geq 1/\beta$ as previously stated? I would think it would have more to do with the error bounds in discretizing the infinite integral in the implementation.

- (line 450) In our streamlined derivation of detailed balance, we have explicitly pointed out that the shifted Metropolis rule is well-defined for $\sigma \ll 1/\beta$, but may cause unnecessary idling (e.g., suppressing transitions near $\omega = 0$). The condition $\sigma \geq 1/\beta$ is therefore a natural scale for σ , as we explain under the "Exact detailed balance with finite uncertainty" paragraph in Methods.

Line 334: Is the reason that the closed form implies algorithmic efficiency that the exponential decay in t allows you to truncate the infinite integral? And does the quasi-locality require a Lieb-Robinson bound? If so, this should probably be explicit.

- (line 487) Yes, the exponential decay in t is indeed responsible for the efficiency and locality, and we have added some more explanation to make this explicit.

Line 358: Does the relationship between N and ϵ implicitly use the fact that the width σ is chosen to be $1/\beta$?

- (line 548) The referee is right, and we have now updated the scaling of N to include the full dependence on σ .

Line 374: You should probably remind the reader that Fig 3A is a simplified, less-efficient Lindbladian simulation algorithm and that you will discuss the more efficient one later.

- We have added a paragraph after Theorem 1 to make this point explicit (the last paragraph before the "Mixing times - numerics" subsection).

Line 400: Theorem 2 doesn't give any indication of the dependence of the running time on ϵ . There is at least a linear factor in $1/\epsilon$, right? Also, the variable c used in the theorem is the same c from the size of the second register first used on page 18, but it seems to be a free-floating variable, and there's no indication here about what constraints govern the choice of the parameter c .

- (line 592) The dependence on epsilon is logarithmic, and now we explain the notation explicitly, that $\tilde{O}(T)$ stands for $O(T \text{ polylog}(T+1/\epsilon))$, and mention the role of c (now renamed q), explaining that $c=\log(2M)$ suffices in typical (Pauli) scenarios.

I think I would have preferred a final theorem statement that gives the running time for the specific Lindbladian from (9) as a function of running time t and error ϵ , using only a black-box algorithm to simulate the dynamics of the Hamiltonian e^{iHt} for one unit of time. At the end of the day, that's what you want to do, right?

- We have included the t -dependence in Theorem 1. The dependence on ϵ is polylogarithmic and is inherited from the general-purpose Lindblad simulation algorithm (Theorem 2). While we have expanded the explicit poly-logarithmic dependence on ϵ right after Theorem 2, we expect that future advances in Lindbladian simulation will further improve this dependence.

Sincerely,
Chi-Fang Chen
on behalf of all authors